



# Sensitivity of formaldehyde (HCHO) column measurements from a geostationary satellite to aerosol temporal variation in East Asia

Hyeong-Ahn Kwon[1], Rokjin J. Park[1], Jaein I. Jeong[1], Seungun Lee[1], Gonzalo González Abad[2], Thomas P. Kurosu[3], Paul I. Palmer[4], and Kelly Chance[2]

[1]School of Earth and Environmental Science, Seoul National University, Seoul, Republic of Korea
[2]Harvard-Smithsonian Center for Astrophysics, Cambridge, Massachusetts, USA
[3]Jet Propulsion Laboratory, Pasadena, California, USA
[4]School of GeoSciences, University of Edinburgh, Edinburgh, UK

*Correspondence to*: Rokjin J. Park (rjpark@snu.ac.kr)

**Abstract.** We examine upcoming geostationary satellite observations of formaldehyde (HCHO) columns in East Asia and the retrieval sensitivity to the temporal variation of air mass factor (AMF) considering the presence of aerosols. Observation system simulation experiments (OSSE) were conducted using a combination of a global 3-D chemical transport model (GEOS-Chem), a radiative transfer model (VLIDORT), and a HCHO retrieval
algorithm developed for Geostationary Environment Monitoring Spectrometer (GEMS), which will be launched in 2019. Application of the retrieval algorithm to simulated hourly radiances yields the retrieved HCHO column concentrations, which are then compared with the GEOS-Chem HCHO columns as a true value for the evaluation of the retrieval algorithm. In order to examine the retrieval sensitivity to the temporal variation of AMF, we compare the
retrieved HCHO columns using monthly versus hourly AMF values and find that the HCHO vertical columns with hourly AMF are in better agreement with the true values, relative to those with monthly AMF. The differences between hourly and monthly AMF range from -0.70 to 0.73 in absolute value and are mainly caused by temporal changes of aerosol chemical composition: scattering aerosol enhances AMF, whereas absorbing aerosol reduces it. The
temporal variations of AMF caused by aerosols increase and decrease HCHO VCDs by 84% and 34%, respectively, compared to HCHO VCDs using monthly AMF. We apply our calculated AMF with the aerosol effects to OMI HCHO products in March, 2006 when Asian





dust storms occurred and find -18% – 33% changes in the retrieved HCHO columns in East Asia. The impact of aerosols cannot be neglected for future geostationary observations.

## 1 Introduction

Formaldehyde (HCHO) is mainly produced by the oxidation of hydrocarbons with minor direct emissions from fuel combustion, vegetation, and biomass burning (DiGangi et al., 2012). Because of its short atmospheric lifetime (~1.5 hours) (De Smedt et al., 2008), HCHO vertical columns from satellite measurements have effectively been used to provide constraints on its precursor emissions, especially for biogenic isoprene emissions (Palmer et al., 2003; Abbot et al., 2003; Shim et al., 2005; Fu et al., 2007; Marais et al., 2012), the oxidation of which is the largest natural source of HCHO globally. Zhu et al. (2014) also used temporal oversampling of satellite observed HCHO columns to provide information for anthropogenic non-methane volatile organic compounds (NMVOCs) emissions in eastern Texas.

In East Asia, anthropogenic emissions have dramatically increased owing to the rapid economic growth over the recent decades (Jeong and Park, 2013). Satellite observed HCHO columns show an increasing trend in most East Asian countries, implying the increase of hydrocarbon emissions (De Smedt et al., 2010). On the other hand, Stavrakou et al. (2014) used top-down isoprene emissions constrained by satellite observations to show the decreasing trend of inferred isoprene emissions in China since 2007, caused by decrease of annual temperatures. However, quantification of precursor emissions and its change is extremely challenging and provides large uncertainty in present air quality models in East Asia (Fu et al., 2007). Constraints based on observations, including satellite HCHO column, are thus necessary to better quantify the emission of NMVOCs and its effects on air quality and climate in East Asia.

Column measurements of HCHO from space started in 1995 with the launch of the GOME instrument onboard ERS-2 (Chance et al., 2000). Since then, successive instruments including SCIAMACHY (Wittrock et al., 2006), OMI (Kurosu et al., 2004; González Abad et al., 2015), GOME-2 (De Smedt et al., 2012), and OMPS (Li et al., 2015; González Abad et al., 2016) onboard sun-synchronous satellites have observed global HCHO column concentrations with re-visiting frequencies of 1 to 6 days. Their minimum ground pixel sizes have reduced from $40 \times 320 \ km^2$ (GOME) to $13 \times 24 \ km^2$ (OMI). Accordingly, HCHO global





observations have increased in use to provide observational constraints on biogenic
NMVOCs emissions over the United States (Abbot et al., 2003; Palmer et al., 2003; Palmer et
al., 2006), Europe (Dufour et al., 2009), Asia (Fu et al., 2007; Stavrakou et al., 2014), and
other regions (Barkley et al., 2013; Marais et al., 2012), despite the fact that measurements
from sun-synchronous satellites have limited observation frequency of at most once or twice
a day to once a week for the region of interest. For anthropogenic emissions, the use of
satellite observations for constraining anthropogenic emission is relatively limited because of
lower anthropogenic HCHO concentration relative to biogenic HCHO (Zhu et al., 2014).

In order to overcome the limitations of sun-synchronous satellites and monitor air
quality changes with higher temporal frequency over East Asia, the Korean Ministry of
Environment will launch a geostationary satellite (GEO-KOMPSAT 2B) carrying the
Geostationary Environment Monitoring Spectrometer (GEMS) in 2019. The GEMS
instrument has a spatial resolution of $7 \times 8$ km$^2$ over Seoul, Korea and can measure trace
gases and aerosols every hour during the daytime (at least 8 times a day). Frequent
observations on a finer spatial resolution provide more data with less cloud contamination
compared to those of the sun-synchronous satellites. Sentinel-4 and Tropospheric Emissions
Monitoring of Pollution (TEMPO) missions (Zoogman et al., 2016) for environmental
geostationary satellites in Europe and the United States, respectively, are also in preparation.
Along with those geostationary satellites, the GEMS instrument plays a role in monitoring air
quality changes over East Asia and intercontinental transport of trace gases and aerosols from
source to receptor regions.

Satellite HCHO column observations are sensitive to the changes of the atmospheric
conditions. In particular, the air mass factor (AMF), which is required to convert slant
column densities (SCDs) to vertical column densities (VCDs), depends on cloud properties,
vertical profiles of HCHO, surface reflectance, aerosols, and observation geometry (solar and
viewing zenith angles) (Palmer et al., 2001; Martin et al., 2002; Lee et al., 2009). Gonzi et al.
(2011) examined the sensitivity of AMF to the injection height and optical properties of
aerosols for biomass burning emission constraints using HCHO satellite measurements.
Leitão et al. (2010) examined the aerosol effect on AMF calculation for satellite NO$_2$
observations.

For sun-synchronous satellites, pre-calculated monthly averaged AMF values have been
applied for computational efficiency (De Smedt et al., 2008; González Abad et al., 2015).
With geostationary satellites, however, we are interested in monitoring the diurnal variation





of trace gases and aerosols for which atmospheric conditions can change over the measurement period. Thus, the application of monthly AMF to hourly observations will cause errors in the retrieved column concentrations. As we show below, consideration of hourly AMF values is necessary for geostationary satellite observations.

Here we examine the necessity of hourly AMF values for geostationary satellite observations by analysing the retrieval sensitivity to the temporal variation of AMF considering the hourly variation of aerosols. We calculate AMF values using different temporal variations of atmospheric constituents and quantify retrieval errors given different temporal resolution of AMF values by comparing the retrieved versus true HCHO VCDs in

observation system simulation experiments (OSSE).

## 2 Observation System Simulation Experiments (OSSE)

We conduct the OSSE as illustrated in Fig. 1, using a global 3D chemical transport model (GEOS-Chem) (Bey et al., 2001), the Vector Linearized Discrete Ordinate Radiative Transfer (VLIDORT) model (Spurr, 2006), and a retrieval algorithm developed for GEMS in

this study (Chance et al., 2000; González Abad et al., 2015). Detailed information on GEOS-Chem and VLIDORT can be found in the aforementioned references. Here we briefly discuss our application.

We first perform a global simulation to obtain spatial and temporal distributions of gas and aerosol species using GEOS-Chem v9-01-02. The model is driven by Modern-Era

Retrospective Analysis for Research and Applications (MERRA) and the Goddard Earth Observing System (GEOS-5) reanalysis meteorological data for years 2006 and 2009, respectively. GEOS-Chem has a $2^{\circ} \times 2.5^{\circ}$ (latitude $\times$ longitude) spatial resolution and 47 levels from the surface to 0.01 hPa. Biogenic emission of isoprene is computed using the Model of Emissions of Gases and Aerosols from Nature (MEGAN) version 2.1 (Guenther et

al., 2006). Anthropogenic emissions are taken from the Emissions Database for Global Atmospheric Research (EDGAR) version 2.0 inventory (Olivier et al., 1996) for the globe in a mosaic fashion with the Intercontinental Chemical Transport Experiment Phase B (INTEX-B) inventory developed by Zhang et al. (2009) for Asia. We use monthly biomass burning emissions from the Global Fire Emissions Database (GFED) version 3 inventory (van der

Werf et al., 2010).





All the simulated concentrations of gases and aerosols are archived every hour for the East Asia domain (70-150ºE, 4ºS-54ºN) and are provided as input for other model calculations. For example, aerosol optical properties, which are important input for radiative transport model simulations below, are calculated using Flexible Aerosol Optical Depth

(FlexAOD) with the simulated aerosol concentrations including sulfate-nitrate-ammonium, organic carbon, black carbon, sea salt, and dust aerosols (Hess et al., 1998; Mishchenko et al., 1999; Sinyuk et al., 2003). Hourly aerosol optical depth (AOD), single scattering albedo (SSA), and asymmetry factor are also archived over the domain for the use in radiative transfer calculations.

We then conduct a radiative transport model simulation using VLIDORT driven by the simulated profiles of gases and aerosol optical properties described above as well as meteorological data. We calculate radiances at the top of the atmosphere. The calculated radiances in 300-500 nm spectral range of GEMS with a 0.2 nm spectral sampling become synthetic radiances to simulate GEMS measurements and are referred to as "observed

radiances" henceforth. We use the observed radiances to evaluate the retrieval algorithm and to examine its sensitivity to several parameters. However, the observed radiances do not include any noise terms such as polarization errors and temperature errors of sensors and are not convoluted with a slit function whose information is not available yet. The evaluation of our retrieval algorithm sensitivity and the impact of AMF on HCHO retrievals we derive

below have therefore to be considered a "best case scenario". The radiative transport simulation accounts for the extinction of aerosols and gases including $O_3$, $NO_2$, $SO_2$, and HCHO. Aerosol optical properties at 300 nm, 400 nm, 600 nm, 999 nm are used in the simulation. VLIDORT also yields derivatives of radiances with respect to optical thicknesses of interfering gases that are used to calculate AMF.

Finally, we apply our retrieval algorithm to the observed radiances to obtain the satellite observed HCHO columns. This retrieval process begins by fitting a simple Lambert-Beer model that explains the absorption of trace gases and the scattering by molecules in the atmosphere to the observed radiances by using a non-linear least square method (Chance et al., 2000).

HCHO absorption is so weak that the accuracy of retrievals is very sensitive to the fitting window selections (Hewson et al., 2013). The HCHO absorption bands overlap the $O_3$ absorption bands, which are the strongest interference in the HCHO retrieval, so the fitting window must be selected to minimize the impact of the strong $O_3$ absorption region.




Instruments such as GOME, SCIAMACHY, OMI, and GOME-2 have used slightly different fitting windows. In this study, we select 327.5-358.0 nm for the fitting window of the HCHO retrieval. In the retrieval algorithm, we consider Ring effect (Chance and Spurr, 1997), $O_3$ absorption cross sections at 228 K and 273 K (Daumont et al., 1992; Malicet et al., 1995),

$NO_2$ absorption cross sections at 220 K (Vandaele et al., 1998), $SO_2$ absorption cross sections at 298 K (Hermans et al., 2009; Vandaele et al., 2009), and HCHO absorption cross sections at 300 K (Chance and Orphal, 2011).

For the retrieval of SCDs of target species from sun-synchronous satellites measurements, differential optical absorption spectroscopy (DOAS) method has frequently

been used with a linearized equation of the logarithm of the Lambert-Beer model divided by the solar irradiance ($I_0$) (De Smedt et al., 2008). In this study, we apply the fitting method developed by Chance et al. (2000) that uses the Lambert-Beer model in its original, non-linearized form, which is known to produce smaller fitting residuals than those of DOAS in $O_3$ retrievals at high solar zenith angles (Van Roozendael et al., 2012). GEMS will perform a

measurement every hour during daytime and often observe at high solar zenith angles. In this regard, the non-linearized fitting method may have an advantage over the DOAS method for the radiance fitting with lower residuals at high solar zenith angles. However, rigorous comparisons are also required for geostationary satellite measurements.

SCDs from radiance fitting are converted to vertical amounts considering the path of

solar radiance and viewing geometry of satellites. AMF is a correction factor of the path length of light from the SCDs to the VCDs, including the varying sensitivity of the observations at different altitudes. It is defined as the ratio of the SCDs to the VCDs. Palmer et al. (2001) derived a simple formulation of AMF including scattering and absorption of gases with the vertical integration of a function multiplying scattering weights and vertical

shape factors. The decoupling of the scattering weights and vertical shape factors has the advantage to allow the calculation of them separate using a radiative transfer model and a chemical transport model, respectively. In this study, AMF calculation is conducted using Eq. (1) below in VLIDORT simulations with hourly trace gases profiles including HCHO and aerosol optical properties from GEOS-Chem.

$$AMF = -\frac{1}{\int k_\lambda \rho \, dz} \int \frac{\partial \ln I}{\partial \tau} d\tau \qquad (1)$$




, where $k_\lambda$ indicates the absorption cross section (cm$^2$ molecule$^{-1}$) at each wavelength, $\rho$ is a number density (molecules cm$^{-3}$), $\tau$ is an optical thickness, and $I$ is a radiance. We use AMF values at 346 nm, which is in the middle of the HCHO fitting window.

### 3 Evaluation of the HCHO retrieval algorithm

In this section, we evaluate the HCHO retrieval algorithm developed for GEMS using the OSSE discussed in Sect. 2. The simulated data, including trace gases (O$_3$, NO$_2$, SO$_2$, and HCHO) concentrations, meteorological data, and aerosol optical properties for March, June, September, and December 2006, are used to calculate radiances in the OSSE as explained above. In radiance calculations, solar zenith angles are used at 11 local standard time (LST)

of Seoul on the equinoxes and solstices (21$^{st}$ of each month), and viewing zenith angles are calculated based on GEMS orbit at ~36,000 km altitude above ~128.2$^o$E longitude at the equator. We assume a lambertian surface reflectance of 0.05. As mentioned above, the simulated radiances do not include noise and errors. SCDs retrieved by radiance fitting are converted to VCDs using AMF with and without aerosols. Previous algorithms used in sun-

synchronous satellites to retrieve HCHO have not accounted for aerosol effects on AMF calculations.

Figure 2 presents GEOS-Chem HCHO VCDs in East Asia (1$^{st}$ column) used in the OSSE to compute the observed radiances. Highest GEOS-Chem HCHO columns occur in Southeast Asia including the Indochina Peninsula and Indonesia mainly driven by large

biomass burning emissions, whose seasonal variations slightly differ depending on the regions. Values in the Indochina Peninsula (92-105$^o$E, 12-25$^o$N) are highest in March-May, which is a typical dry season. In Indonesia (100-118$^o$E, 2$^o$S-4$^o$N), HCHO columns are generally high throughout the whole year because of the biogenic emissions in tropical forests. In 2006, a strong El Niño occurred and resulted in massive fire events in Borneo and

Sumatra for September-October (Stavrakou et al., 2009), which led to enhancements of HCHO columns up to $4.3 \times 10^{16}$ molecules cm$^{-2}$ in September. On the other hand, seasonal variability at mid-latitudes (> 25$^o$N) follows those of biogenic activity. For example, HCHO VCDs in China (105-120$^o$E, 25-40$^o$N) increase to $1.3 \times 10^{16}$ molecules cm$^{-2}$ in June and September but decrease to $4.6 \times 10^{15}$ and $3.7 \times 10^{15}$ molecules cm$^{-2}$ in March and

December, respectively.





Retrieved HCHO VCDs are also presented in Fig. 2. Most HCHO VCDs measured on sun-synchronous satellites including OMI and GOME-2 have been retrieved without considering the aerosol effect on AMF because cloud retrieval algorithms are coupled with the presence of aerosols (De Smedt et al., 2008; González Abad et al., 2015). In order to

avoid complexity and to understand the retrieval sensitivity to the presence of aerosols in East Asia, we only focus on clear sky conditions and compare a retrieval using AMF with aerosols to that using AMF without aerosols. Retrieved HCHO VCDs accounting for aerosols ($2^{nd}$ column in Fig. 2) show spatial and seasonal patterns similar to GEOS-Chem values. Coefficients of determination ($R^2$) between the retrieved and simulated HCHO VCDs for

each month are 0.98 or higher with regression slopes close to one (0.95-1.01) except for winter ($R^2 = 0.95$, slope = 1.05). This is due to the limited capability of our algorithm at high solar zenith angle and low HCHO concentrations. For the calculation of regression coefficients, we exclude grids over 88.4° solar zenith angle in winter (upper left corner in the domain) due to the high bias arising from high solar and viewing zenith angle.

Results retrieved using no aerosols ($3^{rd}$ column in Fig. 2) also show a similar spatial and seasonal variation but with a high bias with respect to the values retrieved using aerosols and GEOS-Chem. We find differences (HCHO VCDs with – without aerosols) are generally negative over China and India. The presence of aerosols in AMF appears to result in the decreases of HCHO columns up to 20% in regions where aerosol concentrations are high

such as China, India, and biomass burning areas. In biogenic emission regions, the effects of biogenic aerosols on AMF are negligible except for biomass burning cases occurring over Indonesia in September and Indochina in March. However, HCHO VCDs are also increased by 14% due to aerosols in regions with high solar and viewing zenith angles.

In radiance fitting, the averaged root mean square (RMS) error of fitting residuals is

$3.3 \times 10^{-4}$, and averaged HCHO slant column error is $1.9 \times 10^{15}$ molecules cm$^{-2}$. Both are relatively small, indicating a successful retrieval because no additional errors are included in the observed radiances. Our retrieved values should be considered as the best-case retrievals that we can obtain from the satellite observations. More detailed error analysis is beyond the scope of this study and will be conducted as soon as the GEMS instrument parameters are

available. We generally find that fitting RMS errors and HCHO slant column errors tend to depend on solar and viewing zenith angles so that these errors gradually increase in regions further away from the position of sun and satellite. HCHO slant column errors also depend on





HCHO concentration in the atmosphere, and uncertainties decrease to $8.1 \times 10^{14}$ molecules cm$^{-2}$ in regions with intense wildfires in March when HCHO concentrations are very high.

## 4 Sensitivity of the HCHO retrieval to aerosol temporal variations

Aerosol concentrations in East Asia are high because of natural and anthropogenic
contributions. Sources include soil dust aerosols from deserts and arid regions predominant in
spring, black carbon and organic aerosols from biomass burning, and inorganic sulfate-
nitrate-ammonium (SNA) aerosols from industrial activities caused by rapid economic
development. In particular, natural aerosols such as dust and biomass burning aerosols are
transported to the free troposphere by mechanisms such as frontal passages or thermally
driven convection associated with their formation processes. Aerosol layers over the polluted
boundary layer can play a role in modulating incoming and backscattered radiance and thus
cause an error in the retrieved quantities of satellite measurements. In order to correct this
error, we need to consider the effect of aerosols on measured radiances. In this section, we
investigate different effects of aerosols when measuring HCHO columns from GEMS by
including aerosols in AMF calculations and further examining the retrieval sensitivity with
respect to temporal variation of aerosol optical properties and HCHO profiles.

Gonzi et al. (2011) showed AMF changes to AOD and aerosol profiles with different
SSA (0.8, 0.95). In Gonzi et al. (2011), aerosol layers aloft resulted in higher HCHO VCDs
caused by decreasing AMF as AOD increases regardless of aerosol optical properties.
However, that AMF including scattering aerosols in the boundary layer tends to increase as
AOD increases. Our AMF calculation is consistent with the previous study. AMF is sensitive
to AOD but changes with different SSA. Increasing AOD for scattering aerosols ($\omega = 0.92$)
results in an increase of AMF whereas the absorbing aerosols ($\omega = 0.82$) result in a decrease
of AMF. As the peak altitude of aerosols increases from the surface to 2 km, AMF sharply
decreases but remains relatively the same above 2 km as expected. This indicates that the
aerosol height may not be a significant factor for GEMS HCHO measurements with a fully
developed planetary boundary layer during the afternoon, but could be an important
consideration with a shallow boundary layer, a residual aerosol layer above, and long range
transport of aerosols.
For geostationary satellites, temporal variations of atmospheric conditions such as
HCHO profiles and aerosols can affect AMF calculations. We use the OSSE described in





Sect. 2 to examine temporal variations of AMF values and the impact on HCHO retrievals. We first compute hourly AMF values using GEOS-Chem simulations for June 2009 and monthly AMF using average simulated data for the whole month. Then, hourly versus monthly AMF values are applied to obtain retrieved HCHO VCDs. Figure 3 compares

HCHO VCDs simulated by GEOS-Chem and retrieved VCDs with monthly and hourly AMF at 346 nm at 11, 12, and 13 LST in Seoul in 21 June 2009. We take the model results as true values in the comparison with the retrieved HCHO VCDs.

GEOS-Chem simulation shows high HCHO VCDs of $1.2 \times 10^{16}$ molecules cm$^{-2}$ over Indonesia near the equator, reflecting high biogenic emissions from tropical forests.

Enhanced HCHO VCDs as high as $9.6 \times 10^{15}$ molecules cm$^{-2}$ over northern Indochina peninsula and China (100-120ºE, 20-35ºN) result from biogenic and anthropogenic emissions. HCHO VCDs do not change over time. We find that the retrieved HCHO VCDs with monthly and hourly AMF are generally consistent with the model results, reproducing spatial distributions of HCHO VCDs. However, HCHO VCDs retrieved with hourly AMF

have better agreement with GEOS-Chem than those retrieved using monthly AMF especially over China. Retrieved HCHO columns using monthly AMF are generally biased high, compared to the others over China.

Figure 4 shows a scatterplot comparison of retrieved VCDs versus model simulations averaged for 3 hours (11-13 LST) over East Asia (105-135ºE, 15-45ºN). We find no

significant biases in the two retrieved products compared with the true values; regression slopes are close to one for both. However, the coefficients of determination ($R^2$) between the retrieved versus true VCDs differ significantly and are 0.75 and 0.99 for the retrieved VCDs with monthly and hourly AMF, respectively, indicating a better performance of the retrieval using hourly AMF relative to that with monthly AMF.

This discrepancy between the two retrieved products is caused by temporal variations of HCHO vertical profiles and aerosols, which result in the changes of AMF values. First column in Fig. 5 shows the difference between hourly (AMF$_h$) and monthly AMF (AMF$_m$) and individual contributions of HCHO profiles and aerosols to the difference averaged for 11-13 LST of Seoul on 21 June 2009. First of all, we find that AMF$_h$ is smaller by 0.70 in

absolute value than AMF$_m$ over northern China, while the former is higher up to 0.73 relative to the latter in the middle of China. First column in Fig. 5(b) and 5(c) show the effects of temporal variation of HCHO profiles and aerosols on AMF calculation, respectively. In order to quantify individual contributions to AMFs, each one of the HCHO profiles and aerosol



optical properties is allowed to vary hourly while other variables are kept fixed using monthly averaged data. We find that HCHO profile variations affect AMF over the entire domain. Changes range from -0.45 to 0.37. More pronounced differences shown in China correlate significantly with the effect of aerosols, whose optical properties change with time,

resulting in AMF variations of -0.60 to 0.75. The significant changes occurring in east China with high aerosol loadings show a different sign that the increase is in the south, whereas the decrease occurs in the north.

We also calculate the ratio of $AMF_m$ to $AMF_h$ (2nd column in Fig. 5) that indicates changes of HCHO VCDs with $AMF_h$ relative to those with $AMF_m$. HCHO VCDs using $AMF_h$

is 2.2 times higher and 0.61 times lower than those using $AMF_m$ over China. Contributions of temporal variation of HCHO profiles and aerosols result in the increment of 46% and 84% and the decrement of 25% and 37% to HCHO VCDs using monthly AMF over China, respectively. Martin et al. (2003) and Lee et al. (2009) showed that the aerosol correction factors, which is defined by the ratio of AMF with aerosol to AMF without aerosol, could

vary from 0.7 to 1.15 depending on aerosol chemical composition; AMF increases with scattering aerosols but decreases with absorbing aerosols. Our ratio reflecting temporal variation effects shows a higher sensitivity of HCHO retrieval than that from previous studies.

In order to further understand the factors for the spatial pattern of changes, we compare

hourly AOD and SSA at 300 nm with monthly values (Fig. 6). In general, the region where hourly AOD is larger than monthly AOD corresponds to the region with the significant change of AMF. We find that hourly SSA is lower in northern China and a bit higher in the middle of China than monthly SSA. In other words, absorbing aerosols in northern China result in the decrease of AMF, whereas scattering aerosols in the middle of China cause the

increase of AMF. These spatial patterns of SSA and thus AMF changes are mainly determined by slightly absorbing dust aerosols in the north and mainly scattering inorganic SNA aerosols in the middle as shown in Fig. 6 (c) and (d). However, AMF decreases near Mongolia despite scattering aerosols, representing that aerosols are suspended out of the boundary layer as mentioned above.

Our illustrative results indicate that aerosols and their chemical compositions in East Asia can vary rapidly and may have significant impacts on retrieved HCHO columns. Therefore, use of monthly AMF may cause considerable errors for geostationary satellites



measurements such as GEMS in East Asia. To improve HCHO GEMS retrievals AMF calculations have to consider the diurnal variability of aerosols and its chemical composition.

We note that the performance of our retrieval approach should be considered as a "best case scenario" due to the absence of any noise components in the OSSE studies. Actual
GEMS measurements will contain noise from polarization, temperature fluctuations of the GEMS instrument, stray light, and other sources, which will reduce retrieval sensitivity. However, despite this expected reduction in retrieval sensitivity, the main results on the impact of aerosols from this study will not change fundamentally. In the next section we demonstrate these effects on the real-life example of the OMI HCHO retrievals.

**5 Effects of aerosols on OMI HCHO products**

AMF application for converting SCDs to VCDs of OMI HCHO has been conducted based on a look-up table approach with no consideration of aerosols (González Abad et al., 2015). Here, we apply AMF values with aerosols to OMI HCHO SCDs to examine the effect of aerosols in clear sky conditions (cloud fraction < 0.05) on the retrieved HCHO VCDs
focusing on East Asia in 2006. Cloud fraction included in OMI HCHO products is used, which is provided from OMCLDO2 products (Stammes et al., 2008). The AMF calculation has been conducted similarly using the GEOS-Chem simulations for 2006 with the same anthropogenic emissions of 2009. In order to apply efficiently our values to the OMI SCDs we compute an AMF table as a function of longitude, latitude, AODs (0.1, 0.5, 1.0, 1.5, 2.0),
and SSAs (0.82, 0.87, 0.92, 0.97) at 30º solar zenith angle and 0.0001º viewing zenith angle, which correspond to a nadir overpass in East Asia.

Figure 7 shows monthly averaged AOD and SSA at 354 nm (cloud fraction < 0.05) from OMI UV radiances (OMAERUV) for March 2006. OMAERUV products are derived from measured reflectance from OMI and climatological surface albedo from TOMS at 354 and
388 nm, aerosol type, and aerosol layer height (Torres et al., 2013). Ahn et al. (2014) evaluated AOD from OMAERUV with Aerosol Robotic Network (AERONET) and showed a root mean square error of 0.16 and a correlation coefficient of 0.81 at 44 global sites over 4 years (2005-2008). SSA from OMAERUV shows difference of ±0.03 (±0.05) compared to that of AERONET at 47% (69%) of 269 sites (Jethva et al., 2014). Although Torres et al.
(2013) excluded pixels related to cloud contamination using scene reflectivity and surface reflectance at 388 nm, aerosol index, and aerosol type, we select pixels where cloud fraction



is less than 0.05 to explain explicit effects of aerosols in AMF calculation. High AOD extending from the Taklamakan desert with a relatively low SSA indicates slightly absorbing dust aerosols in East Asia.

We use the OMI aerosol products with AMF table to calculate AMF values. Figure 8(a)
shows the ratio of AMF without to with aerosols, reflecting the effect of aerosols on retrieved HCHO VCDs. Values are mostly less than one, indicating the decrease of HCHO VCDs because of aerosols except over the Gobi desert. Peak decrease is up to 10%. Figure 8(b) represents difference between a mean of hourly AMF using aerosol optical properties at each time and monthly averaged AMF using averaged AOD and SSA for a month, which reflects
the non-linear property of AMF including aerosol and effects from temporal variation of aerosols in AMF. Negative values indicate lower hourly AMF than monthly AMF due to temporal variation of aerosols so that HCHO column concentrations using hourly AMF are higher than those with monthly AMF.

In order to explore different situations, we examine a dust storm event in 23-29 March
2006. AOD and SSA ($1^{st}$ and $2^{nd}$ rows in Fig. 9) are high and relatively low corresponding to dust aerosols transported from the Taklamakan and Gobi deserts. As expected, the ratio of AMF without ($AMF_{no}$) to with aerosols ($AMF_a$) increases during the dust storm ($3^{rd}$ row of Fig. 9). It is a consequence of the absorbing dust aerosols transported by the dust storm. The effects are pronounced over central and northeastern China and are sometimes extended to
downwind regions of Korea and East Sea on 25 and 27 March. The ratio also increases due to biomass burning in Indochina peninsula. The aerosol effects on AMF make HCHO VCDs increase by 26% due to absorbing aerosols and decrease by 29% due to scattering aerosols compared to those using AMF without aerosols.

The effects of temporal variation of aerosols on AMF ($4^{th}$ row in Fig. 9) are mostly
represented over special events such as dust storms and biomass burning. The differences between hourly and monthly AMF using OMI aerosol products range from -0.31 to 0.31 and result in HCHO changes of -18% to 33%. That indicates that aerosol optical properties simultaneously measured for geostationary satellites can be used to calculate AMF for HCHO VCDs and improve HCHO VCDs. However, aerosol layer height is also important to
determine AMF as mentioned in Sect. 4, and the effect on AMF could be considered using simultaneous measurements of aerosol height and model simulations for air quality forecast.



## 6 Summary

Formaldehyde (HCHO) in the atmosphere is a primary oxidation product of NMVOCs. HCHO measurements on sun-synchronous satellites have been used to update the emission inventories of NMVOCs especially for biogenic sources. However, observations from sun-
synchronous satellites are limited, at most once or twice a day, depending on geographic latitude, and are even further constrained by clouds. In order to overcome these disadvantages and monitor air quality, a constellation of geostationary satellites will be launched over Asia, Europe, the United States. In East Asia, the GEMS instrument onboard GEO-KOMPSAT 2B will measure trace gases and aerosols every hour during daytime. Therefore, application of
monthly AMF to convert SCD to VCD may lead to errors in the retrieved HCHO column concentrations due to AMF depending on atmospheric conditions and observation geometry.

In this study we have examined the sensitivity of retrieved HCHO vertical columns to the temporal variation of AMF. We used different temporal AMFs and compared retrieved HCHO VCDs with true values in the OSSE. Two retrieved VCDs with hourly and monthly
AMF were consistent with the true values, but the retrieved HCHO VCDs using hourly AMF showed a better agreement with the true than those using monthly AMF over China. The differences resulted from the temporal changes of aerosols and their chemical compositions in our AMF calculation: scattering aerosols increase AMFs by up to 0.73 and absorbing aerosols decrease them by up to 0.70. The ratio of monthly AMF to hourly AMF, which
reflects changes of HCHO VCDs with hourly AMF in comparison with those with monthly AMF, ranges from 0.62 to 2.2. This retrieval sensitivity was found to be higher than aerosol effects on AMF estimated in previous studies (Martin et al., 2003; Lee et al., 2009). It must be noted that this estimate constitutes a "best case scenario" due to the absence of any noise term in our GEMS radiance simulations.

We also applied an AMF look-up table accounting for the presence of aerosols to OMI HCHO SCDs in order to examine aerosol effects on OMI retrieval primarily focusing on clear sky conditions (cloud fraction < 0.05). We found that the consideration of aerosol effects resulted in a decrease of HCHO VCDs by 10% on a monthly mean basis. In a dust storm event for 23-29 March 2006, the consideration of aerosols for AMF calculation
changed HCHO VCDs from -29% to 26% relative to the original OMI HCHO VCDs. In addition, differences between hourly and monthly AMF were -0.31 to 0.31, and the resulting HCHO VCD changes were -18% to 33%. Our test with the OMI products indicated a





possibility that simultaneously measured aerosol products can be used to calculate AMF considering aerosol and its temporal variation effects to improve the accuracy of HCHO VCDs retrieval.

In this study, we selected pixels in clear sky conditions to examine explicit aerosol

effects on AMF calculation because the retrieval algorithms of aerosol and cloud interact with each other. We may need to investigate interaction effects between aerosol and cloud on AMF when we consider cloud products from satellites to calculate AMF. Finally, an aerosol height in the boundary layer is an important factor in AMF, the effects that can be included by using simultaneously measured aerosol height and model simulations for air quality forecast.

**Acknowledgements**

This work was supported by the Korean Ministry of Environment as part of the Eco-Innovation Project.

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





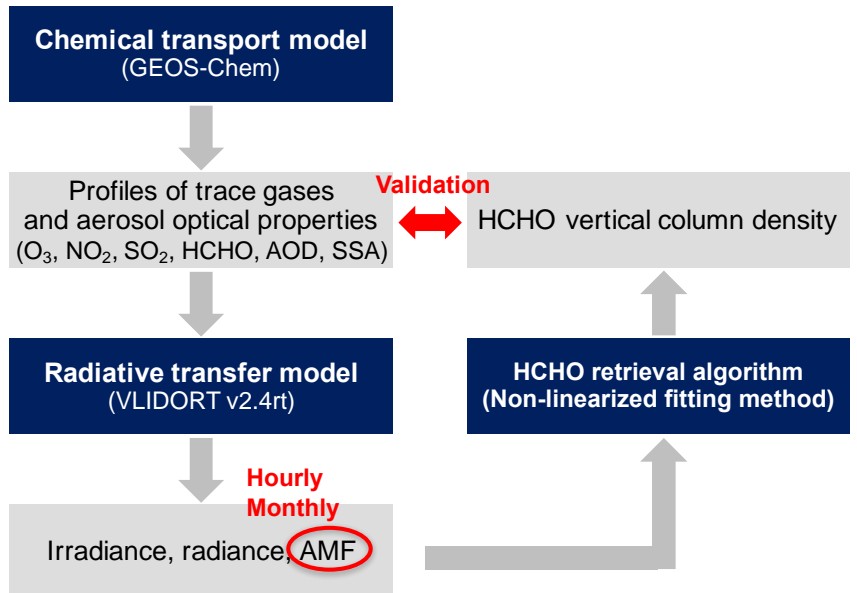

**Figure 1. Schematic diagram of observation system simulation experiments (OSSE) used to validate our retrieval algorithm and to examine its sensitivity to the temporal variation of AMF values. GEOS-Chem driven by the assimilated meteorological data is used to produce profiles of atmospheric constituent concentrations. VLIDORT**
5  **calculates observed radiances measured by geostationary satellites using atmospheric constituent concentrations and meteorological conditions from GEOS-Chem simulations. HCHO retrieval algorithm is developed based on least-squares fitting of a non-linearized Lambert-Beer model and is validated by comparisons between simulated and retrieved column densities of HCHO. The latter is obtained by applying the retrieval algorithm to the observed radiances from VLIDORT. Details are provided in the text.**





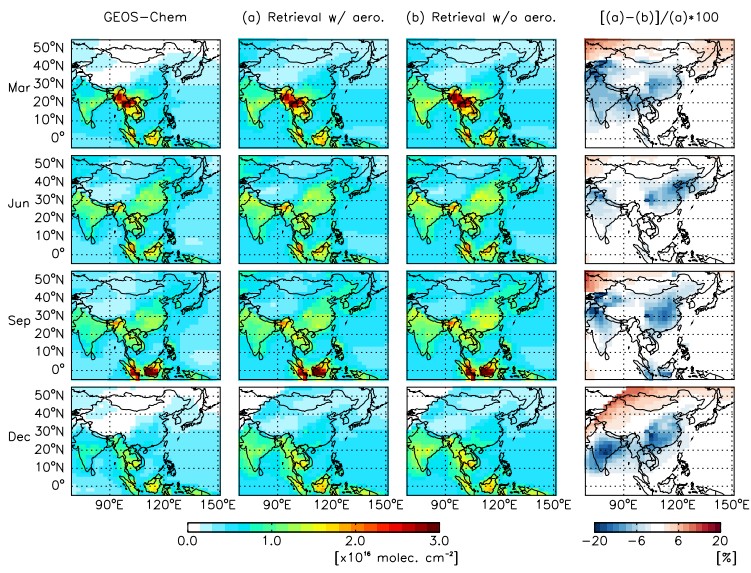

**Figure 2. HCHO vertical column densities (VCDs) simulated from GEOS-Chem (1st column) and retrieved HCHO VCDs using AMF with aerosols (2nd column) and without aerosols (3rd column) for a month of each season in 2006. Relative differences between the two retrievals using AMF with and without aerosols are shown on the 4th column and represent the aerosol effect on the retrieved HCHO VCDs.**



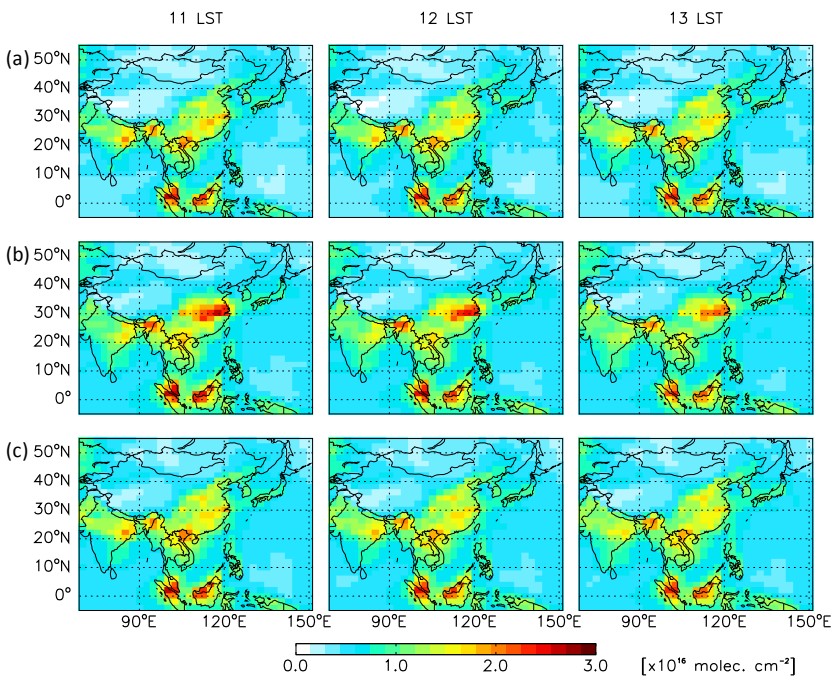

**Figure 3. (a) HCHO VCDs simulated by GEOS-Chem at 11, 12, 13 local standard time (LST) of Seoul on 21 June 2009. (b) Retrieved HCHO VCDs with monthly AMF. (c) Retrieved HCHO VCDs with hourly AMF.**




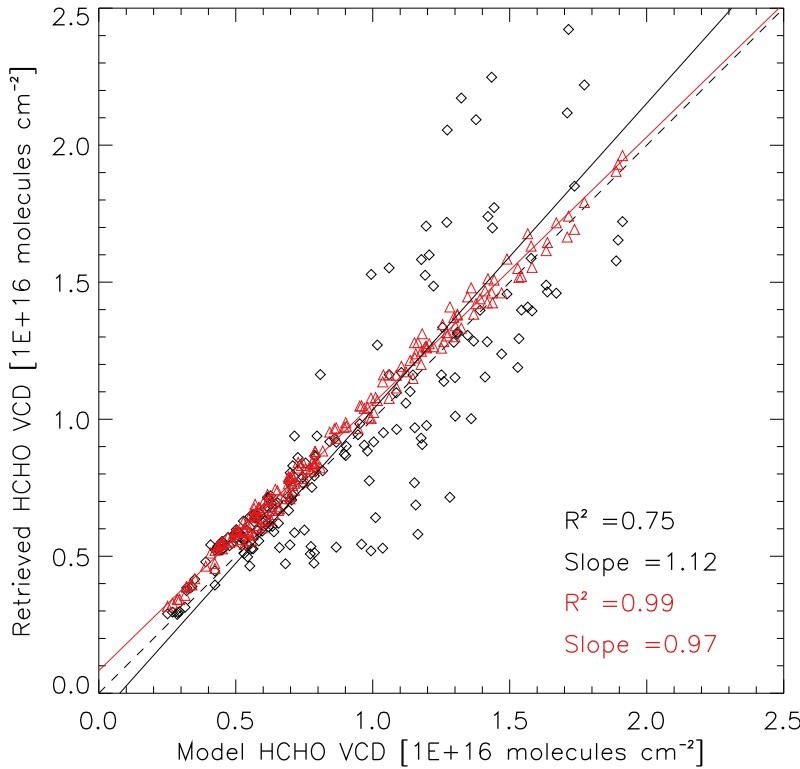

**Figure 4. Comparison of retrieved versus simulated VCDs shown in Fig. 3 over East Asia (105-135ºE, 15-45ºN). Black diamonds and red triangles indicate the retrieved VCDs using monthly and hourly AMF values, respectively. Statistics are also shown as inset.**





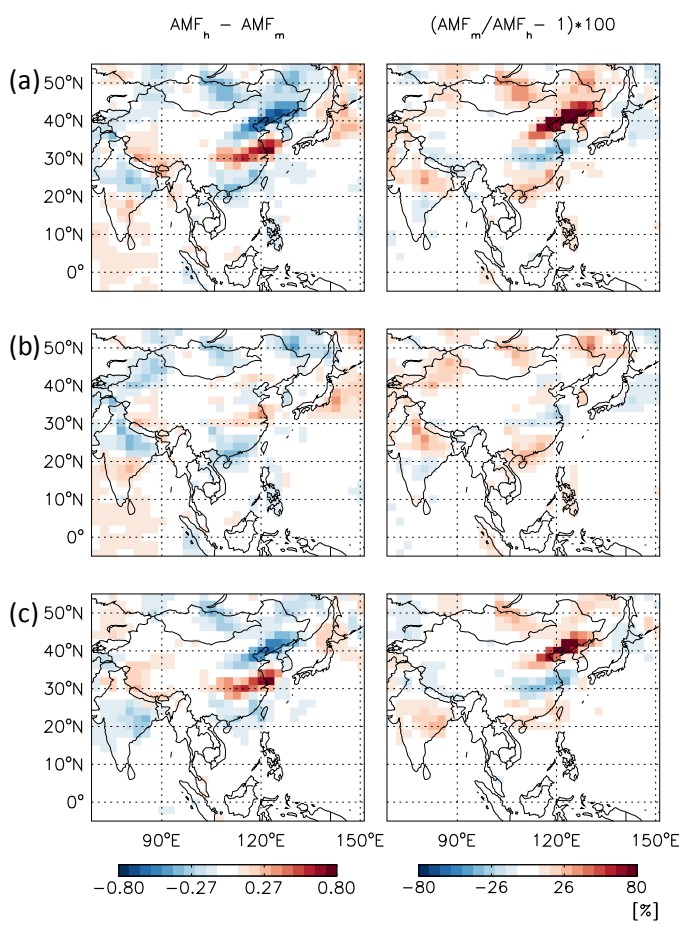

**Figure 5. (a) Differences between hourly and monthly AMF values and relative contributions to them by the temporal changes of (b) HCHO profiles and (c) aerosols for 11-13 LST of Seoul on 21 June 2009.**





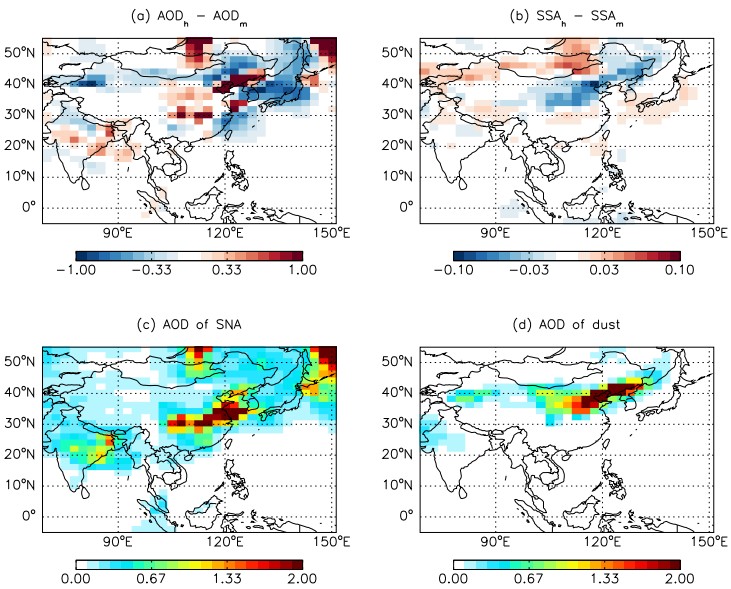

**Figure 6. Averaged differences for 3 hours (11-13 LST on 21 June 2009) between hourly and monthly (a) AOD and (b) SSA. Averaged AOD of (c) sulfate-nitrate-ammonium (SNA) aerosols and (d) soil dust aerosols for 3 hours**





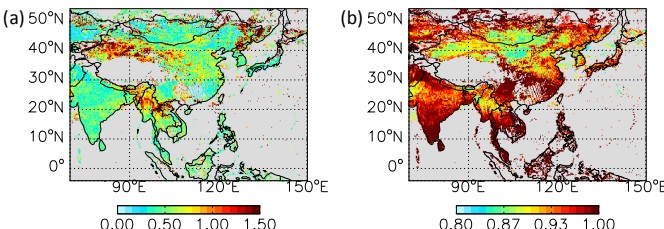

**Figure 7. (a) AOD and (b) SSA at 354 nm from OMI used in AMF calculation for March 2006 in clear sky conditions (cloud fraction < 0.05).**

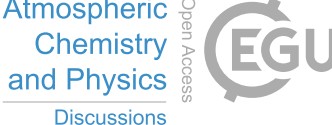



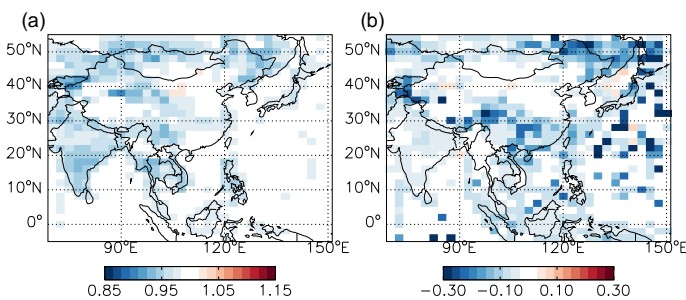

**Figure 8. (a) Ratio of AMFs without aerosols to AMFs with aerosols. It represents the ratio of HCHO VCDs with aerosols to those without aerosols. (b) Differences of monthly AMFs values calculated by taking an average of hourly AMFs values versus calculating monthly AMFs with monthly mean AOD and SSA. Aerosol optical properties used in the calculation are from OMI observations (OMAERUV) for March 2006.**





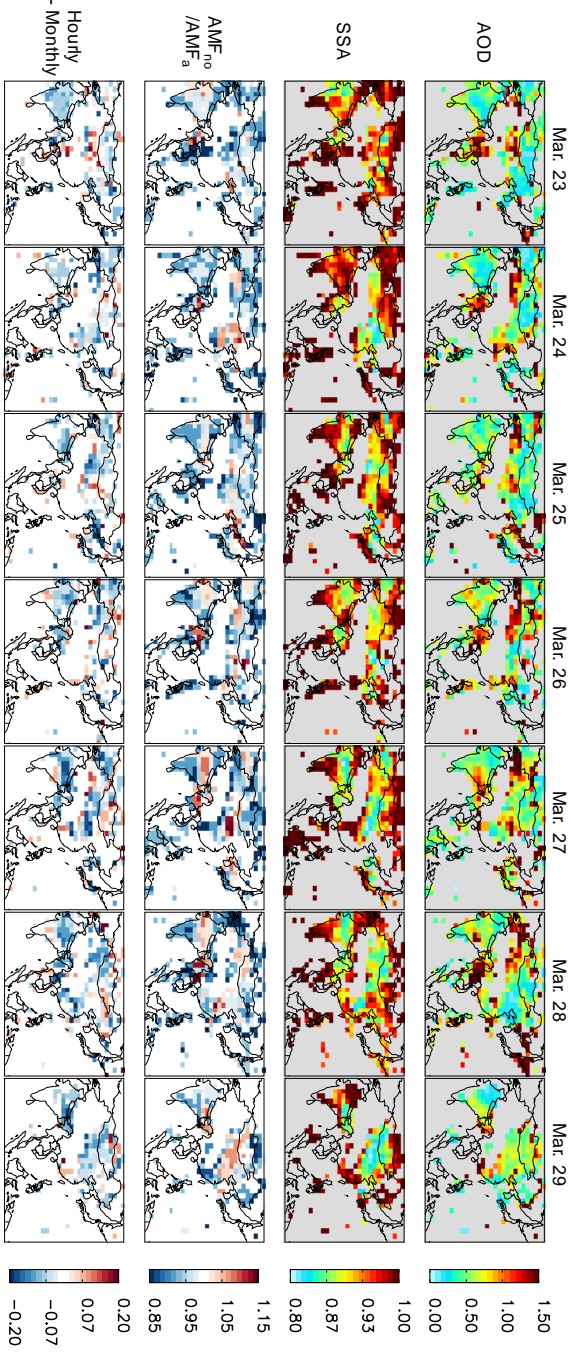

**Figure 9. Values of AOD, SSA, the ratio of AMF values (AMF$_{no}$/AMF$_a$), and differences between hourly and monthly AMF for March 23-29, 2006, when a strong dust event occurred in East Asia. AMF$_{no}$ and AMF$_a$ indicate values without and with aerosols, respectively.**