# Peer review of "Sensitivity of formaldehyde (HCHO) column measurements from a geostationary satellite to temporal variation of air mass factor in East Asia"

_Atmospheric Chemistry and Physics, 2016_

## Referee Comment (RC1) · Anonymous Referee #2 · 2 Nov 2016

**General comments**

This manuscript summarises the results of an OSSE study with the goal to evaluate the sensitivity of HCHO retrievals from a Geostationary sensor (the Korean GEMS mission) to aerosol variability. The OSSE is performed as a best-case scenario, *i.e.* without noise, and tries to identify the best strategy when taking into account explicit aerosol correction into the computation of AMF: either by considering hourly of monthly aerosol variability. This is an important question since trace gas retrievals from a geostationary sensor will deliver an unprecedented high number of observations with a high temporal frequency. This is an interesting study, focusing on a very important topic. In general, this manuscript is well written, but several explanations or sentences should be further explained (please see details in "Technical corrections").

However, I recommend some major revisions to be addressed by the authors before publishing the manuscript. Indeed, in my personal opinion, several elements are either critically missing or not correctly addressed. I would like to emphasize, in particular, the point 1) in "Specific comments" which, I believe, is the most crucial one that needs to be carefully clarified in order to better understand / evaluate the importance of the outcome of this study. As this may cause quite some works, I am open to simple exercises suggested by the authors to answer the questions below.

My main remarks about this manuscript are summarised below, but please see details in "Specific comments":

- The importance of the temporal variability of the aerosol altitude (or vertical distribution), in addition to the optical properties (AOT and SSA) is not specifically addressed and analysed. Some lines in the manuscript mention it but in a too ambiguous way to be able to understand how this parameter was considered in this study. By reading the manuscript, I had the feeling that either the aerosol altitude was considered as negligible, and/or it was just ignored for unknown reasons. By experience, aerosol vertical profile is one of the key parameter when retrieving trace gases from space (if not the most crucial one) and its impact should not be neglected or minimized compared to AOT and/or SSA.
- No analyses focussing on observation times that are only accessible by a geostationary instrument are shown here (*i.e.* outside of the range 11:00-12:00-13:00, typical overpass times of present day polar orbiting satellites). It is a bit a pity since this is one of the original element provided by a geostationary platform.
- The definition of "monthly averaged AMF" and the methodology of computing the averaged profiles are a bit misleading, in particular for readers outside of the HCHO retrieval community.
- The assumed aerosol profile in the OMI HCHO exercise (Section 5 of the manuscript) is not discussed. In addition, it would be nice to compare hourly and monthly HCHO AMF for the Dust storm event of 23-29 March 2009. Only the AMFs with and without aerosols are here compared.

**Specific comments**

1) Aerosol altitude and/or profile?

I struggle to understand the assumptions made by the authors about the shape of vertical distribution of aerosols from hour to hour, day-to-day and month-to-month and how they explicitly impact the computed HCHO AMF depending on the considered methodology (either hourly variability or monthly averages). So far, in my understanding, the authors only considered the impact of assuming constant AOD and SSA properties:

- Line33 P10, "each one of the HCHO profiles and aerosol optical properties is allowed to vary hourly"
- Line19, P11, "we compare hourly AOD and SSA at 300 nm with monthly values"
- Figures 6 and 7 only focus on AOD and SSA variability (which are of course of importance) but do not show the aerosol altitude changes.

These statements and figures, and many others, seem to suggest that the variability of the vertical aerosol profile itself was not explicitly considered, independently and/or combined with their optical property variability. Moreover, the authors mentioned on P.9 that "the peak altitude of aerosols increases from the surface to 2 km". I don't think that such a general statement is always true. Is it a general conclusion supported by referent observations studies over the considered area, or what is seen in the GEOS-Chem model? I would expect to see quite some variations about the height of the peak of the aerosols as it should be strongly driven by 1) the injection height (either in the boundary layer or in the free troposphere), 2) how well the boundary layer (season and synoptic variability) is developed, and 3) specific chemistry processes associated with aerosol particles that may vary depending on their type and the seasons. For example, [Castellanos *et al.*, 2015] demonstrated that biomass burning aerosols extend to high altitudes (about 2 km). But dust particles that are transported over long distance can be found sometimes higher than 2 km. Similarly, sulphate and nitrate particles which result from precursor trace gases may be confined close to the surface where the sources are present.

P9, it is said "Increasing AOD for scattering aerosols (SSA = 0.92) results in an increase of AMF whereas the absorbing aerosols (SSA = 0.82) result in a decrease of AMF". I tend to disagree with such a general statement because:

- Aerosols with SSA=0.92 are still in my view absorbing (although less than with SSA = 0.89). And therefore, I am not sure they can be named "scattering";
- The balance between enhancement or shielding effect strongly depends on 1) the shape of aerosol vertical profile, 2) the shape of trace gas (here HCHO) vertical profile, and thus the relative altitude between the 2 components. Many studies emphasized the importance of the relative vertical distributions of both aerosols and trace gases (such as $NO_2$) on the satellite AMFs [Boersma *et al.*, 2004; Chimot *et al.*, 2016; Shaiganfar *et al.*, 2011; Ma *et al.*, 2013; Kanaya *et al.*, 2014; Wang *et al.*, 2016]. The magnitude then, of the shielding or enhancement effects, relies on the AOD and SSA associated with particles present in the observed scene. Increasing AOD may not always lead to a decrease of AMF, depending on the aerosol altitude and also the surface albedo. For instance, if very scattering particles are located far from the surface and above the tropospheric HCHO bulk, then we should expect to see an increase of enhancement effect with increasing AOD...
- Absorbing aerosols mostly reduce the sensitivity to HCHO concentration [De Smedt *et al.*, 2008] which can result either in a stronger shield effect or a lower enhancement effect compared to scattering particles, depending again on their relative altitude to the HCHO tropospheric bulk.

The authors should give clarifications how much the vertical distribution of aerosols, based on full GEMS-Chem simulations, varies and how the relative altitudes with respect to HCHO vary as well. I trust this information should be available. Is there a dependency from day-to-day or on the seasons?

Furthermore, how the vertical profile of the particles was considered in the present work: was a full vertical profile simulated every hour by GEMS? Or did the authors only consider 1 finite and homogeneous aerosol layer with variable mid-level of pressure / altitude? Of course, I understand that finding a good that finding a good aerosol profile shape estimate is a complex task, but any assumption made about this should be clarified here.

Did the authors average the vertical profiles as well or did they keep them constant hour-to-hour and day-to-day? All these elements are at least as important as hourly AOD, and much more than hourly SSA (as considered in Figure 6 and so), and should have crucial impacts on the variability of HCHO AMFs. I suggest that, in addition of monthly averages of SSA and AOD, the authors indicate us how monthly averages of the vertical profile shape and/or the effective aerosol altitude impact as well the accuracy of the results.

Finally, could the authors clarify and support with figures or references the statement on P. 9, lines 25-28 "This indicates that the aerosol height may not be a significant factor for GEMS HCHO measurements with a fully developed planetary boundary layer height during the afternoon, but could be an important consideration with a shallow boundary layer, a residual aerosol layer above, and long range transport aerosols"? I do not either understand the message of the authors here...

I realize that my demands, here, may cause quite a lot of work for the authors. If they cannot fully be addressed by coupling the transport-chemistry model for aerosol profile shape estimates, I would like the authors to propose then simple aerosol profile shape sensitivity exercises with academic scenarios (*e.g.* low, intermediate and high aerosol profile), to compute the AMF for these scenarios and address the conclusions. If not, then I think that the limitations of this study (*i.e.* one important parameter not considered in the temporal aerosol variability) should be explicitly written in the title, abstract and other places of the manuscript.

2) Notion of "monthly averaged AMF" is ambiguous

The notion of monthly averaged AMF is a little ambiguous. [De Smedt *et al.*, 2008] & [Gonzalez Abad *et al.*, 2015] do not apply a monthly averaged AMF to GOME single pixels but a specific AMF deduced for each observation pixel, based, among other elements:

- A climatology surface albedo [Koelemeijer *et al.*, 2003] which provides monthly Lambert-equivalent reflectivity at 335 nm;
- And monthly vertical profiles of HCHO distribution from a global chemical transport model (GEOS-CHEM or IMAGES).

The other parameters such as effective clouds, angles, surface altitude / pressure are not averaged at the monthly scale but used on a daily basis. Therefore, the mentioned references in this paper did not strictly use a monthly averaged AMF as stated by the author.

Same about the monthly average AMF of the author here: are only aerosols and HCHO profiles averaged or also other parameters? Following point 1) above, what was averaged regarding the aerosols: AOD and SSA only? Or the vertical profile as well? Or was this last element kept constant? I suggest the author to clearly define the monthly average AMF at the beginning of the manuscript.

3) Clarification of monthly average definition?

Following point 2) above, could the authors precise the period over which the averages were computed? Were they performed over all times of all days in 1 month, or were the averages computed over all days at 12:00 only? Are all the times, or only some of them, considered for the monthly averages?

4) Typical geostationary observation times

Why in Section 4 and on figures 3-5 do the authors only show the impact of the different AMFs at 11:00-12:00-13:00? These times are typically encountered by LEO instruments. But with a geostationary sensor, it could be interested to evaluate the impacts outside of this time range such as early in the morning (9:00-11:00) and close to the end of the afternoons (15:00-17:00).

5) OMI HCHO exercise

Following the discussions above, could the authors:

- Detail which altitude and vertical profile they considered when computing the OMI HCHO AMF? Does it come from GEOS-Chem simulations? In my knowledge, the OMI aerosol product from [Torres *et al.*, 2013] includes AOD and SSA but no vertical profiles.
- Regarding the dust storm event of March 2006 from 23 to 29, could the authors show as well the ratio of hourly *vs.* monthly AMF? Only the ratio of AMF without vs. with aerosols is here shown.

6) HCHO aerosol correction AMF

The author mentioned in Section 3 that "previous algorithms used in sun-synchroneous satellites to retrieve HCHO have not accounted for aerosol effects on AMF calculations".

This is not correct. They corrected for aerosol effects but in an implicit way: *i.e.* the effective cloud parameters are used to partially correct these effects since the cloud retrieval algorithm is perturbed over cloud-free scenes but dominated by aerosol particles. These parameters are either derived from the $O_2$-band and/or the $O_2$-$O_2$ band. The authors [De Smedt *et al.*, 2008] and [Gonzales *et al.*, 2015] clearly said "the presence of aerosols is not explicitly accounted for".

Similarly to the other trace gas retrievals from UV-Vis air quality satellite measurements, the use of a simple Lambertian cloud-scheme, although allows to mitigate their impacts, does not apply a comprehensive correction. See [Boersma *et al.*, 2004, 2011; Chimot *et al.*, 2016; Castellanos et al., 2015] who explained this mechanism in case of tropospheric $NO_2$ AMF calculations.

Here, the author considers an explicit aerosol correction scheme on the HCHO AMF computation. The relevant question here is then, what would be the best strategy if an explicit aerosol correction is assumed: monthly average or hourly aerosol profile and properties?

Assuming that the author would not have enough explicit information about aerosol properties and vertical distribution, would the use of daily effective cloud parameters, derived for each single observation pixel, be enough to compensate of temporal variability of aerosol effects?

**Technical corrections**

Abstract:
- 29: Please see my general comments about scattering and absorbing aerosols and correct your general statement accordingly.
- P2, 2: Please precise that you are talking about the impact of aerosol variability, not the aerosols in general.

P2, 30: "frequencies of 1 to 6 days". I suggest to replace by "between 1 and 6 days".

P3, 16: Please add references about Sentinel-4.

P3, 31: "pre-calculated monthly averaged AMF": please precise following point 2) above.

P4, 2-4: these lines are more appropriate in the conclusion section, not in the introduction, since they summarise your results of this manuscript.

P5, 12-15: please reformulate. Computed radiances cannot "become" synthetic radiances…

P5, 21: Were $H_2O$ and $O_2$-$O_2$ included as well?

P6, 30 and equation 1: I do not fully understand how this equation has been derived and did not manage to find it in other references. Could you please provide with 1-2 details about it and any references supporting it? What are the limits of the integrals?

P.9, title of section 4: the sensitivity of the HCHO retrieval to the HCHO profile was investigated too (to be added in the title).

P.9, 4-8: Please add references supporting these statements here (*e.g.* Eck *et al.*, 2005; Jethva *et al.*, 2014)

P.9, 17-29: Please see my major remarks in point 1) above (cf. Details about aerosol altitude and vertical profile), and update this sub-section accordingly.

P.9, 21: "Our AMF calculation is consistent with the previous study". Which study are you referring to? In which sense your AMF is consistent? In terms of precision or employed methodology? Please clarify.

P.9, 30-31: this statement is hard to understand, since the previous lines somehow said that aerosol profiles are not important....Please clarify or reformulate.

P.30, 7-8: Which figure are you referring to?

P.10-11, 30-1: Following point 1) above, please clarify if you kept constant or made vary the aerosol profile? How was this parameter considered here and how did it impact your results?

P.11, 23-25: "In other words, absorbing aerosols [...] cause the increase of AMF": How can you deduce that? Is it always true or should not it depend on the aerosol / HCHO altitude?

P.12, last sub-section of section 4: Not sure if this is necessary here to repeat the explanations about "best case scenario'.

P.13 29-30: "aerosol layer height is also important to determine AMF". I agree but since no analysis w.r.t this parameter are given before, it is quite hard to understand why the authors write this here...Please clarify.

P. 14, 1-11: Please check what is really useful for the conclusion, and not redundant with the general part also present in the introduction. For example, it is not necessary here to repeat the nature of HCHO, why sun-synchroneous satellites are limited etc...
"constellation of geostationary": first time this notion is introduced. Could you please precise it?

P.14, 19: Would the ratio of hourly AMF to monthly AMF not be more useful (than the ratio of monthly to hourly) to illustrate the variability into HCHO VCDs?

P.14, 32-33: "Our test with the OMI products indicated a possibility that simultaneously measured aerosol products can be used to calculate AMF considering aerosol".
This was illustrated based on the OMI AOT and SSA in the UV, but not about the aerosol layer height. Any future expectations regarding this last variable?

P14, 8-10: The authors mentioned the importance of aerosol height in the boundary layer and to use simultaneous measurements. But no measurements about aerosols in the boundary layer are shown and used here. Where could it come from? Are such measurements available somewhere?

P21, Figure 1: Did you compute and use the vertical averaging kernel to convert the GEOS-Chen trace gas profile into vertical column densities in order to validate your retrievals? How do you compute them and where should they be present in your OSSE diagram?

P23, Figure 3: Could you please also times that are available from geostationary observations but not from sensors like OMI (*i.e.* early in the morning, late in the afternoon)?

P24, Figure 4: please indicate for which time(s) of the day are plotted these retrievals.

P25, Figure 5: The sign of the absolute and relative differences are opposite, and thus the colours are reversed between the columns (i.e. what is red on the left, in absolute, becomes blue on the right in relative...). Please correct this.

P28, Figure 8: The ratio of the 2 AMFs is not strictly equal to the ratio of the 2 VDCs, since these last variables include artefacts due to the spectral fit when deriving the slant column densities. However, it represents the part of AMF computation errors included in the VDC products at the end. Please correct your second statement, in the caption, accordingly.

**Additional bibliography recommended**

Boersma, K. F., Eskes, H. J., and Brinksma, E. J.: Error analysis for tropospheric $NO_2$ retrieval from space, J. Geophys. Res., 109, D04311, doi:10.1029/2003JD003962, 2004.

Castellanos, P., Boersma, K. F., Torres, O., and de Haan, J. F.: OMI tropospheric $NO_2$ air mass factors over South America: effects of biomass burning aerosols, Atmospheric Measurement Techniques, 8, 3831–3849, doi:10.5194/amt-8-3831-2015, http://www.atmos-meas-tech.net/8/3831/2015/, 2015.

Chimot, J., Vlemmix, T., Veefkind, J. P., de Haan, J. F., and Levelt, P. F.: Impact of aerosols on the OMI tropospheric $NO_2$ retrievals over industrialized regions: how accurate is the aerosol correction of cloud-free scenes *via* a simple cloud model?, Atmospheric Measurement Techniques, 9, 359–382, doi:10.5194/amt-9-359-2016, http://www.atmos-meas-tech.net/9/359/2016/, 2016.

de Smedt, I., Müller, J.-F., Stavrakou, T., van der A, R., Eskes, H., and Van Roozendael, M.: Twelve years of global observations of formaldehyde in the troposphere using GOME and SCIAMACHY sensors, Atmospheric Chemistry and Physics, 8, 4947–4963, doi:10.5194/acp-8-4947-2008, http://www.atmos-chem-phys.net/8/4947/2008/, 2008.

Ma, J. Z., Beirle, S., Jin, J. L., Shaiganfar, R., Yan, P., and Wagner, T.: Tropospheric $NO_2$ vertical column densities over Beijing: results of the first three years of ground-based MAX-DOAS measurements (2008 and 2011) and satellite validation, Atmospheric Chemistry and Physics, 13, 1547–1567, doi:10.5194/acp-13-1547-2013, http://www.atmos-chem-phys.net/13/1547/2013/, 2013.

Shaiganfar, R., Beirle, S., Sharma, M., Chauhan, A., Singh, R. P., and Wagner, T.: Estimation of NOx emissions from Delhi using Car MAXDOAS observations and comparison with OMI satellite data, Atmospheric Chemistry and Physics, 11, 10 871–10 887, doi:10.5194/acp-11-10871-2011, http://www.atmos-chem-phys.net/11/10871/2011/, 2011.

Wang, Y., Beirle, S., lampel, J., Koukouli, M., De Smedt, I., Theys, N., Li, A., Wu, D., Xie, P., Liu, C., Van Roozendael, M., and Wagner, T.: Validation of OMI, GOME-2A and GOME-2B tropospheric NO2, SO2 and HCHO products using MAX-DOAS observations from 2011 to 2014 in Wuxi, China, Atmos. Chem. Phys. Discuss., doi:10.5194/acp-2016-735, in review, 2016.

---

## Referee Comment (RC2) · Anonymous Referee #1 · 19 Dec 2016

This paper presents an OSSE study to evaluate the retrieval sensitivity of HCHO vertical columns observed by geostationary satellites due to the temporal variations of aerosols. This study is conducted specifically to evaluate the retrieval algorithm for the upcoming Korean geostationary satellite, GEMS. The OSSE was conducted with no noise, such that the resulting retrieval should be considered a "best-case scenario".

The authors found that the temporal variation of aerosols result in significant changes in AMF, leading to errors in HCHO retrieval. Specifically, the effects of aerosols were attributed to the single-scattering albedo, the AOD, and the vertical distribution of aerosols relative to the HCHO, consistent with previous studies. Overall, the authors showed that neglecting the temporal (hourly) variability of aerosols would lead to -37%

to 84% biases in the retrieved HCHO VCD. Thus the impact of aerosols cannot be neglected for future geostationary observations.

The topic is of great interest to the community, as the use of satellite observations of trace gases in air quality studies continue to expand and several geo-stationary satellite instruments are soon coming online. The experiments were thoughtfully designed, and the paper was well written. Personally I only have one major comment that I wish the authors would further address.

My only major suggestion is for the authors to consider the use of monthly mean "hourly" AMF, e.g. monthly mean AMFs for 9AM, 10AM, 11AM local time ... The only reason for using monthly mean AMFs would be to reduce computation such that operational VCD products can be delievered quickly. It seems the use of "monthly mean hourly" AMFs would not only be quick, but also address a lot of the aerosol temporal effects, since for each month, the main diurnal variability of AMFs is driven by the vertical profile of aerosols (which in turn is driven by the development of the PBL). This seems like something the authors can address without too much additional computation.

Minor comment: Page 8, lines 20-21: "In biogenic emission regions, the effects of biogenic aerosols on AMF are negligible . . ."This may be true, but it would be nice to have some quantification. How large is the contribution of biogenic aerosols to total AOD?

---

## Author Comment (AC1) · 29 Jan 2017

Responses to Referee's Comments

We appreciate careful reading and lots of valuable comments.
We wrote referee's comments in black, our responses to comments in blue and italics, and the revised manuscript in red.

**Referee #1:**

My only major suggestion is for the authors to consider the use of monthly mean "hourly" AMF, e.g. monthly mean AMFs for 9AM, 10AM, 11AM local time ... The only reason for using monthly mean AMFs would be to reduce computation such that operational VCD products can be delivered quickly. It seems the use of "monthly mean hourly" AMFs would not only be quick, but also address a lot of the aerosol temporal effects, since for each month, the main diurnal variability of AMFs is driven by the vertical profile of aerosols (which in turn is driven by the development of the PBL). This seems like something the authors can address without too much additional computation.

*Following the reviewer's suggestion, we calculated monthly mean "hourly AMF" ($AMF_{mh}$) and included our discussion on the results using $AMF_{mh}$ in Fig. 3-4 in the revised manuscript as follows:*

**Here, we use three AMF specifications associated with the temporal variation of input data for AMF calculations. Input data include HCHO profiles, aerosol optical properties and profiles, temperatures, pressures, and other interfering gases ($O_3$, $NO_2$, and $SO_2$) from GEOS-Chem simulations. We use monthly, hourly, and monthly-averaged hourly input data at each model grid to compute $AMF_m$, $AMF_h$, and $AMF_{mh}$, respectively, for June 2009.**

**…**

**Figure 4 shows scatterplot comparisons of retrieved VCDs versus model simulations at 9, 12, and 18 LST of Seoul over China (105-120ºE, 15-45ºN). We find some biases in the retrieved products using $AMF_m$ and $AMF_{mh}$ compared with the true values and the results with $AMF_h$. Regression slopes are close to one for the results using**

$AMF_h$ (0.96-1.08) but higher than one for the results using $AMF_m$ (1.14-1.31) and $AMF_{mh}$ (1.08-1.24). The coefficients of determination ($R^2$) between the retrieved versus true VCDs differ significantly and are 0.73, 0.83, and 0.99 for the retrieved VCDs with $AMF_m$, $AMF_{mh}$, and $AMF_h$ at 12 LST, respectively, indicating the best performance of the retrieval using $AMF_h$ relative to those with the other AMFs.

We find that both the regression slope and $R^2$ for the results using $AMF_{mh}$ suggest a better performance than those with $AMF_m$, particularly at 12 LST, but do not show any significant improvement at 9 and 18 LST. We infer from this that the temporal variability of species, caused by the diurnal variation of the planetary boundary layer (PBL), mostly explains the difference between the retrievals using $AMF_m$ and $AMF_{mh}$. Accounting for this diurnal variability appears to be important for the retrieval when the PBL is fully developed and the active chemical processes typically occur. Therefore, we think that the use of $AMF_{mh}$ could be an alternative and more efficient way to improve HCHO VCD retrievals for geostationary satellites with less computation required relative to the use of $AMF_h$.

[Figure]

**Figure 3. (a) HCHO VCDs simulated by GEOS-Chem at 9, 12, and 18 local standard time (LST) of Seoul on 21 June 2009. (b) Retrieved HCHO VCDs with AMF$_m$. (c) Retrieved HCHO VCDs with AMF$_h$. (d) Retrieved HCHO VCDs with AMF$_{mh}$.**

[Figure]

**Figure 4. Comparison of the retrieved versus simulated VCDs shown in Fig. 3 over China (105-120°E, 15-45°N). Black diamonds, red triangles, and blue squares denote the retrieved VCDs using AMF$_m$, AMF$_h$, and AMF$_{mh}$, respectively. Statistics are shown as insets.**

Minor comment: Page 8, lines 20-21: "In biogenic emission regions, the effects of biogenic aerosols on AMF are negligible . . ."This may be true, but it would be nice to

have some quantification. How large is the contribution of biogenic aerosols to total AOD?

*We clarified the sentences as follows*:

**In biogenic emission regions, AOD at 300 nm is low (<0.1) and thus its effect of AMF is relatively minor except for biomass burning cases occurring over Indonesia (100-120$^{\circ}$E, 4$^{\circ}$S-5$^{\circ}$N) in September and Indochina (100-120$^{\circ}$E, 10-20$^{\circ}$N) in March.**

---

## Author Comment (AC2) · 29 Jan 2017

Responses to Referee's Comments

We appreciate careful reading and lots of valuable comments.

We wrote referee's comments in black, our responses to comments in blue and italics, and
the revised manuscript in red.

**Referee #2:**

Specific comments

1) Aerosol altitude and/or profile?

I struggle to understand the assumptions made by the authors about the shape of vertical
distribution of aerosols from hour to hour, day-to-day and month-to-month and how they
explicitly impact the computed HCHO AMF depending on the considered methodology
(either hourly variability or monthly averages). So far, in my understanding, the authors
only considered the impact of assuming constant AOD and SSA properties:

- ♦ Line33 P10, "each one of the HCHO profiles and aerosol optical properties is
allowed to vary hourly"
- ♦ Line19, P11, "we compare hourly AOD and SSA at 300 nm with monthly values"
- ♦ Figures 6 and 7 only focus on AOD and SSA variability (which are of course of
importance) but do not show the aerosol altitude changes.

These statements and figures, and many others, seem to suggest that the variability of the
vertical aerosol profile itself was not explicitly considered, independently and/or
combined with their optical property variability.

*We understand that we were not clear enough about our method to conduct the
sensitivity test of AMF calculations to the temporal variation of aerosol optical
properties. To compute hourly AMF values, we used hourly simulations of gas and
aerosol concentrations as well as meteorological data. The effect of aerosol altitude
variation was already included in our study, but we did not separate this effect from the
overall aerosol effects. In the revised manuscript, we separately quantify the temporal
variation effects of aerosol vertical profile and aerosol optical properties (AOD and
SSA). The detailed description is included in Sec. 4 with Fig. 5 in the revised*

*manuscript as well as in our responses below.*

Moreover, the authors mentioned on P.9 that "the peak altitude of aerosols increases from the surface to 2 km". I don't think that such a general statement is always true. Is it a general conclusion supported by referent observations studies over the considered area, or what is seen in the GEOS-Chem model? I would expect to see quite some variations about the height of the peak of the aerosols as it should be strongly driven by 1) the injection height (either in the boundary layer or in the free troposphere), 2) how well the boundary layer (season and synoptic variability) is developed, and 3) specific chemistry processes associated with aerosol particles that may vary depending on their type and the seasons. For example, [Castellanos et al., 2015] demonstrated that biomass burning aerosols extend to high altitudes (about 2 km). But dust particles that are transported over long distance can be found sometimes higher than 2 km. Similarly, sulphate and nitrate particles which result from precursor trace gases may be confined close to the surface where the sources are present.

*Yes, we agree with you. We removed that general statement in the revised manuscript and included the description for the effect of aerosol altitude change on AMF calculation. We also conducted a new sensitivity study of the temporal variation of aerosol altitude separately and discussed it in the revised paper as follow:*

**We also find that aerosol profile variation is important for the AMF calculation as well as aerosol optical property. That is evident, in particular, over the middle of eastern China where the increment of AMF occurs owing to HCHO above aerosol layers (Fig. 5(d)). The resulting change of AMF is consistent with the study by Chimot et al. (2016) that suggested an enhancement (albedo) effect associated with the relative distribution between HCHO and aerosol. The enhancement effect refers to the increased HCHO absorption within and above aerosol layers because of an increased photon path length caused by aerosol backscatter (Chimot et al., 2016).**

[Figure]

**Figure 5. (a) Differences between $AMF_h$ and $AMF_m$ values and relative contributions to them by the temporal changes of (b) HCHO profiles, (c) aerosol optical properties, and (d) aerosol vertical distributions. The first to third columns are results at 9, 12, and 18 LST at Seoul on 21 June 2009. The fourth column gives percentage differences for the ratio of $AMF_m$ to $AMF_h$ indicating changes of HCHO VCDs with $AMF_h$ relative to those with $AMF_m$ at 12 LST.**

P9, it is said "Increasing AOD for scattering aerosols (SSA = 0.92) results in an increase of AMF whereas the absorbing aerosols (SSA = 0.82) result in a decrease of AMF". I tend to disagree with such a general statement because:

♦ Aerosols with SSA=0.92 are still in my view absorbing (although less than with SSA = 0.89). And therefore, I am not sure they can be named "scattering";

*We agree with you and revised our manuscript significantly for clarity.*

♦ The balance between enhancement or shielding effect strongly depends on 1) the shape of aerosol vertical profile, 2) the shape of trace gas (here HCHO) vertical profile, and thus the relative altitude between the 2 components. Many studies emphasized the importance of the relative vertical distributions of both aerosols

and trace gases (such as NO2) on the satellite AMFs [Boersma et al., 2004; Chimot et al., 2016; Shaiganfar et al., 2011; Ma et al., 2013; Kanaya et al., 2014; Wang et al., 2016]. The magnitude then, of the shielding or enhancement effects, relies on the AOD and SSA associated with particles present in the observed scene. Increasing AOD may not always lead to a decrease of AMF, depending on the aerosol altitude and also the surface albedo. For instance, if very scattering particles are located far from the surface and above the tropospheric HCHO bulk, then we should expect to see an increase of enhancement effect with increasing AOD...

♦ Absorbing aerosols mostly reduce the sensitivity to HCHO concentration [De Smedt et al., 2008] which can result either in a stronger shield effect or a lower enhancement effect compared to scattering particles, depending again on their relative altitude to the HCHO tropospheric bulk.

*Thanks for the constructive comment. Following your comment, we conducted the new sensitivity explained above to clarify the dependency of aerosol profiles on AMF calculation in the revised manuscript. The results are shown in Fig. 5 with our discussion above.*

*In addition, we cited previous study related with the dependency of relative distribution between HCHO and aerosol on AMF calculation.*

The authors should give clarifications how much the vertical distribution of aerosols, based on full GEMS-Chem simulations, varies and how the relative altitudes with respect to HCHO vary as well. I trust this information should be available. Is there a dependency from day-to-day or on the seasons?

*As shown in our response to your first comment, all the data used for AMF calculation are from GEOS-Chem, which simulates hourly variation of aerosols and gases in East Asia. Detailed computation of how the vertical distribution of aerosols and HCHO change would be a bit cumbersome, although the information is available as you indicated. Instead, we showed in Fig. 5 in the revised manuscript the temporal variation effects of HCHO and aerosol vertical distributions on AMF calculations in East Asia.*

*Figures 5(b) and (d) also show HCHO and aerosols vertical shapes effects on AMF compared to AMF using monthly averaged HCHO and aerosol profiles, respectively.*

*To make it clear to understand aerosol profile effects, we compared aerosol profiles (solid) at 12 LST with monthly mean aerosol profiles (dashed) over eastern China representing significant AMF changes. Blue lines indicate aerosol profiles over the northeastern China, where $AMF_h$ is lower than $AMF_m$ at 12 LST. Red lines denote aerosol profiles over the middle of eastern China, where $AMF_h$ is higher than $AMF_m$. As we discussed above, in the middle of eastern China (red lines), aerosols are more distributed near the surface compared to monthly mean aerosol profiles (Fig. S1), resulting in an enhancement effect and the increment of AMF. In the northeastern China (blue lines), aerosols are aloft above 2 km so that we expect a shielding effect resulting in the decrement of AMF values. However, $AMF_h$ did not decrease significantly due to aerosol profile effects on AMF calculation in Fig 5(d). That is because monthly mean SSA used for the quantification of aerosol profile effects is higher than SSA at 12 LST, shown in Fig. 6 (b) of the manuscript. Shielding effects for scattering aerosols could be relatively weaker than those of absorbing aerosols because multiple scattering of aerosols increases a possibility for HCHO to absorb photons.*

[Figure]

*Figure S1. AOD profiles over the eastern China representing pronounced AMF changes in Fig. 5. Solid and dashed lines indicate AOD profiles at 12 LST and monthly mean AOD profiles. Blue and red colors indicate over the regions where AMF values decrease and increase, respectively.*

Furthermore, how the vertical profile of the particles was considered in the present work: was a full vertical profile simulated every hour by GEMS? Or did the authors only consider 1 finite and homogeneous aerosol layer with variable mid-level of pressure / altitude? Of course, I understand that finding a good that finding a good aerosol profile shape estimate is a complex task, but any assumption made about this should be clarified here.

*We used hourly aerosols simulated from GEOS-Chem. For the sensitivity studies, our AMF calculation is described as follows:*

**We use the OSSE described in Sect. 2 to examine AMF temporal variations and their impact on HCHO retrievals. For geostationary satellites, temporal changes of atmospheric conditions can affect AMF calculations. Here, we use three AMF specifications associated with the temporal variation of input data for AMF calculations. Input data include HCHO profiles, aerosol optical properties and profiles, temperatures, pressures, and other interfering gases ($O_3$, $NO_2$, and $SO_2$) from GEOS-Chem simulations. We use monthly, hourly, and monthly-averaged hourly input data at each model grid to compute $AMF_m$, $AMF_h$, and $AMF_{mh}$, respectively, for June 2009. First of all, all the three AMFs vary hourly as functions of the solar zenith angle and location. However, at a given solar zenith angle and location, $AMF_m$ does not change due to use of monthly mean input dataset over all times of all days in a given month, $AMF_h$ changes every hour within a month, and $AMF_{mh}$ changes hourly with no day-to-day variation. Then, we apply $AMF_m$, $AMF_h$, and $AMF_{mh}$ to retrieved HCHO SCDs in order to obtain retrieved HCHO VCDs.**

*However, in order to make AMF table in Sect. 5, we used aerosol profiles, AOD and SSA, HCHO profiles, and other parameters monthly averaged for March 2006. Although relative altitude between aerosols and HCHO is important, we cannot use aerosol layer heights from OMI for now. Therefore, we made AMF table as a function*

*of AOD and SSA only. If an aerosol layer height is retrieved from GEMS or other satellites (Park et al., 2016), we should include aerosol heights in AMF table.*

*We clarified usage of monthly data for AMF table in the section of "Effects of aerosols on OMI HCHO products".*

**The AMF calculation has been conducted similarly with monthly mean data from the GEOS-Chem simulations for 2006. … An aerosol layer height is also important to determine AMF as discussed in Sect. 4. However, the information is not yet available from the satellites with ultraviolet and visible channels so that our AMF look-up table is not a function of aerosol layer heights.**

Did the authors average the vertical profiles as well or did they keep them constant hour-to-hour and day-to-day? All these elements are at least as important as hourly AOD, and much more than hourly SSA (as considered in Figure 6 and so), and should have crucial impacts on the variability of HCHO AMFs. I suggest that, in addition of monthly averages of SSA and AOD, the authors indicate us how monthly averages of the vertical profile shape and/or the effective aerosol altitude impact as well the accuracy of the results.

*Please see our responses above. We also rewrite our manuscript to clarify this issue as follows:*

**In order to quantify individual contributions to AMF differences between the two, each of the HCHO profiles, aerosol optical properties, and aerosol vertical distributions is allowed to vary hourly while other variables are kept fixed using monthly averaged data for AMF calculation.**

Finally, could the authors clarify and support with figures or references the statement on P. 9, lines 25-28 "This indicates that the aerosol height may not be a significant factor for GEMS HCHO measurements with a fully developed planetary boundary layer height during the afternoon, but could be an important consideration with a shallow boundary layer, a residual aerosol layer above, and long range transport aerosols"? I do not either

understand the message of the authors here...

*As shown in our earlier responses above and in the revised manuscript, the aerosol profile variation is also very important for AMF calculation. We greatly appreciate the reviewer's comment on this issue, which improves our work considerably.*

I realize that my demands, here, may cause quite a lot of work for the authors. If they cannot fully be addressed by coupling the transport-chemistry model for aerosol profile shape estimates, I would like the authors to propose then simple aerosol profile shape sensitivity exercises with academic scenarios (e.g. low, intermediate and high aerosol profile), to compute the AMF for these scenarios and address the conclusions. If not, then I think that the limitations of this study (i.e. one important parameter not considered in the temporal aerosol variability) should be explicitly written in the title, abstract and other places of the manuscript.

*Thanks for the valuable and constructive comment! We think that this comment is quite important not only for our present study but also for future GEMS observations. Therefore, we explicitly quantify the temporal variation effect of both aerosol optical properties and vertical distributions as was discussed above. Our quantification is shown in Fig. 5 using the OSSE and we also cited previous studies to show the importance of relative distributions between aerosols and HCHO for potential readers to understand it clearly in the revised manuscript.*

2) Notion of "monthly averaged AMF" is ambiguous

The notion of monthly averaged AMF is a little ambiguous. [De Smedt et al., 2008] & [Gonzalez Abad et al., 2015] do not apply a monthly averaged AMF to GOME single pixels but a specific AMF deduced for each observation pixel, based, among other elements:

- ♦ A climatology surface albedo [Koelemeijer et al., 2003] which provides monthly Lambert- equivalent reflectivity at 335 nm;
- ♦ And monthly vertical profiles of HCHO distribution from a global chemical transport model (GEOS-CHEM or IMAGES).

The other parameters such as effective clouds, angles, surface altitude / pressure are not

averaged at the monthly scale but used on a daily basis. Therefore, the mentioned references in this paper did not strictly use a monthly averaged AMF as stated by the author.

Same about the monthly average AMF of the author here: are only aerosols and HCHO profiles averaged or also other parameters? Following point 1) above, what was averaged regarding the aerosols: AOD and SSA only? Or the vertical profile as well? Or was this last element kept constant? I suggest the author to clearly define the monthly average AMF at the beginning of the manuscript.

*As you mentioned, definition of monthly averaged AMF is ambiguous. We referred to monthly AMF as AMF calculated using all monthly mean values, including HCHO, aerosol vertical profiles, and AOD and SSA. The line you referred was clarified as follows:*

**For sun-synchronous satellites, pre-calculated AMFs determined by monthly averaged HCHO and aerosol vertical profiles have been applied for computational efficiency (De Smedt et al., 2008; González Abad et al., 2015).**

*We clarified our definition of monthly AMF, hourly AMF, and monthly mean hourly AMF. Please see P.5 18-28 above.*

3) Clarification of monthly average definition?
Following point 2) above, could the authors precise the period over which the averages were computed? Were they performed over all times of all days in 1 month, or were the averages computed over all days at 12:00 only? Are all the times, or only some of them, considered for the monthly averages?

*As we mentioned point 2) above, monthly AMF is calculated using monthly averaged data over all times of all days in the whole month at SZA of each time. Please see our answers in point 2)*

*In addition, we added VCDs using monthly mean "hourly AMF" ($AMF_{mh}$) in Fig. 3*

*and Fig. 4. Corresponding discussion is included in the revised manuscript as follows:*

We find that both the regression slope and $R^2$ for the results using $AMF_{mh}$ suggest a better performance than those with $AMF_m$, particularly at 12 LST, but do not show any significant improvement at 9 and 18 LST. We infer from this that the temporal variability of species, caused by the diurnal variation of the planetary boundary layer (PBL), mostly explains the difference between the retrievals using $AMF_m$ and $AMF_{mh}$. Accounting for this diurnal variability appears to be important for the retrieval when the PBL is fully developed and the active chemical processes typically occur. Therefore, we think that the use of $AMF_{mh}$ could be an alternative and more efficient way to improve HCHO VCD retrievals for geostationary satellites with less computation required relative to the use of $AMF_h$.

[Figure]

**Figure 3. (a) HCHO VCDs simulated by GEOS-Chem at 9, 12, and 18 local standard time (LST) of Seoul on 21 June 2009. (b) Retrieved HCHO VCDs with $AMF_m$. (c) Retrieved HCHO VCDs with $AMF_h$. (d) Retrieved HCHO VCDs with $AMF_{mh}$.**

[Figure]

[Figure]

[Figure]

Figure 4. Comparison of the retrieved versus simulated VCDs shown in Fig. 3 over China (105-120°E, 15-45°N). Black diamonds, red triangles, and blue squares denote the retrieved VCDs using $AMF_m$, $AMF_h$, and $AMF_{mh}$, respectively. Statistics are shown as insets.

4) Typical geostationary observation times

Why in Section 4 and on figures 3-5 do the authors only show the impact of the different AMFs at 11:00-12:00-13:00? These times are typically encountered by LEO instruments. But with a geostationary sensor, it could be interested to evaluate the impacts outside of this time range such as early in the morning (9:00-11:00) and close to the end of the afternoons (15:00-17:00).

*Following your comment, we included our calculations at 9, 12, and 18 LST in Fig. 3, 4, and 5 in the revised manuscript.*

5) OMI HCHO exercise

Following the discussions above, could the authors:

- Detail which altitude and vertical profile they considered when computing the OMI HCHO AMF? Does it come from GEOS-Chem simulations? In my knowledge, the OMI aerosol product from [Torres et al., 2013] includes AOD and SSA but no vertical profiles.

*For AMF table calculations, we used monthly mean vertical profiles from GEOS-Chem, which were averaged for all times of all days in March 2006. OMI aerosol products do not include aerosol layer heights as you indicated, so we examined only AOD and SSA effects on AMF. We revised and clarified sentences related with your comments.*

**Previous AMF applications to convert SCDs to VCDs of OMI HCHO are based on a look-up table approach with no explicit consideration of aerosols (González Abad et al., 2015). Here, we apply AMF values with an explicit consideration of aerosols to OMI HCHO SCDs to examine the effect of aerosol presence and its temporal variation in clear sky conditions (cloud fraction < 0.05) on the retrieved HCHO VCDs focusing on East Asia in 2006. The cloud fraction included in OMI HCHO products is used, which is provided from OMCLDO2 products (Stammes et al., 2008). The AMF calculation has been conducted similarly with monthly mean data from the GEOS-Chem simulations for 2006. In order to apply efficiently our values to the OMI SCDs we compute an AMF look-up table as a function of longitude, latitude, AODs (0.1, 0.5, 1.0, 1.5, 2.0), SSAs (0.82, 0.87, 0.92, 0.97), solar zenith angles (5º, 30º, 60º, 80º), and viewing zenith angles (0º, 10º, 20º, 30º, 40º, 50º, 60º, 70º, 80º). An aerosol layer height is also important to determine AMF as discussed in Sect. 4. However, the information is not yet available from the satellites with ultraviolet and visible channels so that our AMF look-up table is not a function of aerosol layer heights.**

  - Regarding the dust storm event of March 2006 from 23 to 29, could the authors show as well the ratio of hourly vs. monthly AMF? Only the ratio of AMF without vs. with aerosols is here shown.

*We changed a difference between hourly and monthly AMF to the ratio of monthly to hourly AMF reflecting HCHO changes due to the temporal effects. We revised the manuscript as follows.*

**In order to examine aerosol temporal variation effects on AMF calculation, we use the same AMF specifications discussed in Sect. 4. $AMF_h$ denotes AMF using aerosol optical properties at each measurement time, and $AMF_m$ is AMF using monthly mean AOD and SSA.**

**…**

**Here we illustrate that the temporal variation effects of AOD and SSA on the AMF calculation (4th row in Fig. 9) can adequately be accounted for using satellite**

**observations especially for episodic events such as dust storms and biomass burning. $AMF_m$ uses OMI monthly mean AOD and SSA for March 2006, and $AMF_h$ uses them at each measurement time. The ratio of $AMF_m$ to $AMF_h$ ranges from 0.68 to 1.47 reflecting HCHO changes of -32% to 47% by using $AMF_h$ compared to VCDs with $AMF_m$. That indicates that aerosol optical properties simultaneously measured for geostationary satellites can be used to calculate AMF for HCHO VCDs and to reduce the associated uncertainty with the retrieved products.**

[Figure]

**Figure 9. Values of AOD, SSA, aerosol optical property effects on AMF ($AMF_{no}/AMF_a$), and temporal effects of aerosol optical properties on AMF ($AMF_m/AMF_h$) for March 23-29, 2006, when a strong dust event occurred in East Asia. $AMF_{no}$ and $AMF_a$ indicate values without and with aerosols, respectively. $AMF_m$ is a value using monthly mean AOD and SSA from OMI. $AMF_h$ is a value using AOD and SSA from OMI at each measurement time.**

6) HCHO aerosol correction AMF

The author mentioned in Section 3 that "previous algorithms used in sun-synchroneous satellites to retrieve HCHO have not accounted for aerosol effects on AMF calculations". This is not correct. They corrected for aerosol effects but in an implicit way: i.e. the effective cloud parameters are used to partially correct these effects since the cloud retrieval algorithm is perturbed over cloud-free scenes but dominated by aerosol particles. These parameters are either derived from the O2-band and/or the O2-O2 band. The authors [De Smedt et al., 2008] and [Gonzales et al., 2015] clearly said "the presence of aerosols is not explicitly accounted for".

Similarly to the other trace gas retrievals from UV-Vis air quality satellite measurements, the use of a simple Lambertian cloud-scheme, although allows to mitigate their impacts, does not apply a comprehensive correction. See [Boersma et al., 2004, 2011; Chimot et al., 2016; Castellanos et al., 2015] who explained this mechanism in case of tropospheric NO2 AMF calculations.

*We agree with you. We clarified the sentence as follows.*

**Most HCHO VCDs for previous sun-synchronous satellites including OMI and GOME-2 have been retrieved without the explicit consideration of aerosol effects on AMFs because aerosols are implicitly accounted for from satellite cloud products, which are coupled with the presence of aerosols (De Smedt et al., 2008; González Abad et al., 2015).**

Here, the author considers an explicit aerosol correction scheme on the HCHO AMF computation. The relevant question here is then, what would be the best strategy if an explicit aerosol correction is assumed: monthly average or hourly aerosol profile and properties?

*We included our suggestion in the revised manuscript as follows:*

**Therefore, we think that the use of $AMF_{mh}$ could be an alternative and more efficient way to improve HCHO VCD retrievals for geostationary satellites with less computation required relative to the use of $AMF_h$.**

Assuming that the author would not have enough explicit information about aerosol properties and vertical distribution, would the use of daily effective cloud parameters, derived for each single observation pixel, be enough to compensate of temporal variability of aerosol effects?

*Thanks for the suggestion and we consider it in our future study.*

Technical corrections

Abstract:

• 29: Please see my general comments about scattering and absorbing aerosols and correct your general statement accordingly.

5      *We removed the sentences.*

• P2, 2: Please precise that you are talking about the impact of aerosol variability, not the aerosols in general.

10      *We changed "the impact of aerosols" to "the impact of aerosol variability" in the revised manuscript.*

P2, 30: "frequencies of 1 to 6 days". I suggest to replace by "between 1 and 6 days".

15      *Yes, we changed it.*

P3, 16: Please add references about Sentinel-4.

       *We added the reference:*

       **Ingmann, P., Veihelmann, B., Langen, J., Lamarre, D., Stark, H., and Courrèges-Lacoste, G. B.: Requirements for the GMES atmosphere service and ESA's implementation concept: Sentinels-4/-5 and-5p, Remote Sens. Environ., 120, 58–69, doi:10.1016/j.rse.2012.01.023, 2012.**

P3, 31: "pre-calculated monthly averaged AMF": please precise following point 2) above.

       *We rewrote the sentence as follows:*

30      **For sun-synchronous satellites, pre-calculated AMFs determined by monthly averaged HCHO and aerosol vertical profiles have been applied for computational**

**efficiency (De Smedt et al., 2008; González Abad et al., 2015).**

P4, 2-4: these lines are more appropriate in the conclusion section, not in the introduction, since they summarise your results of this manuscript.

*We removed the sentences following your suggestion.*

P5, 12-15: please reformulate. Computed radiances cannot "become" synthetic radiances...

*We modified the sentences as follows:*

**The calculated radiances in 300-500 nm spectral range of GEMS with a 0.2 nm spectral sampling are assumed as synthetic radiances to simulate GEMS measurements**

P5, 21: Were H2O and O2-O2 included as well?

*$H_2O$ is not significant in fitting window (327.5-358 nm) of HCHO, but $O_2$-$O_2$ collision interferes near 350 nm in the fitting window. However, we did not consider $H_2O$ and $O_2$-$O_2$.*

P6, 30 and equation 1: I do not fully understand how this equation has been derived and did not manage to find it in other references. Could you please provide with 1-2 details about it and any references supporting it? What are the limits of the integrals?

*The equation came from Eq. (9) in Palmer et al. (2001). The limits of the integrals ranges from 0 to optical thickness for vertical column.*
*We revised it as follows:*

**We conduct AMF calculations in VLIDORT simulations using Eq. (1) from Palmer**

**et al. (2001) with hourly trace gas profiles including HCHO and aerosol profiles from GEOS-Chem.**

$$AMF = -\frac{1}{\int_0^{TOA} k_\lambda \rho \, dz} \int_0^{\tau_v} \frac{\partial \ln I}{\partial \tau} d\tau, \qquad (1)$$

**where $k_\lambda$ indicates the absorption cross section (cm$^2$ molecule$^{-1}$) at each wavelength, $\rho$ is a number density (molecules cm$^{-3}$), TOA stands for top of the atmosphere, $\tau$ and $\tau_v$ are an optical thickness and that of vertical column, respectively, and $I$ is a radiance. We use AMF values at 346 nm, which is in the middle of the HCHO fitting window.**

P.9, title of section 4: the sensitivity of the HCHO retrieval to the HCHO profile was investigated too (to be added in the title).

*We revised the title as "Sensitivity of the HCHO retrieval to AMF temporal specifications"*

P.9, 4-8: Please add references supporting these statements here (e.g. Eck et al., 2005; Jethva et al., 2014)

*Thanks and we added the references.*

**Eck, T. F., Holben, B. N., Dubovik, O., Smirnov, A., Goloub, P., Chen, H. B., Chatenet, B., Gomes, L., Zhang, X. Y., Tsay, S. C., Ji, Q., Giles, D., and Slutsker, I.: Columnar aerosol optical properties at AERONET sites in central eastern Asia and aerosol transport to the tropical mid-Pacific, J. Geophys. Res.-Atmos., 110, n/a-n/a, 10.1029/2004JD005274, 2005.**
**Jethva, H., Torres, O., and Ahn, C.: Global assessment of OMI aerosol single-scattering albedo using ground-based AERONET inversion, J. Geophys. Res.-Atmos., 119, 9020-9040, 2014.**

P.9, 17-29: Please see my major remarks in point 1) above (cf. Details about aerosol

altitude and vertical profile), and update this sub-section accordingly.

*We answered to your comments about point 1) above.*

5      P.9, 21: "Our AMF calculation is consistent with the previous study". Which study are you referring to? In which sense your AMF is consistent? In terms of precision or employed methodology? Please clarify.

*We removed the general statement related aerosol height in the revised manuscript.*
10    *Instead, we separated temporal variation effects of aerosol profile from overall aerosol effects. Please see revised paragraphs in point 1) and Fig. 5 above.*

P.9, 30-31: this statement is hard to understand, since the previous lines somehow said that aerosol profiles are not important....Please clarify or reformulate.

*We rewrote the paragraphs. Please see P.5 15-17 in this response above.*

P.30, 7-8: Which figure are you referring to?

20    *We clarified the sentence in the revised manuscript.*

**Figure 3 shows that GEOS-Chem simulation has large HCHO VCDs …**

P.10-11, 30-1: Following point 1) above, please clarify if you kept constant or made vary
25    the aerosol profile? How was this parameter considered here and how did it impact your results?

*The individual effects of optical property and profile was quantified in the revised manuscript. We explained the effects of optical property and profile on AMF in point*
30    *1) above. Please see P.2 and Fig. 5.*

P.11, 23-25: "In other words, absorbing aerosols [...] cause the increase of AMF": How can you deduce that? Is it always true or should not it depend on the aerosol / HCHO altitude?

*Temporal effect of optical property was clarified to separate optical property and profile effects from overall aerosol effects. Please see results at 12 LST in Fig. 5(c) above.*

P.12, last sub-section of section 4: Not sure if this is necessary here to repeat the explanations about "best case scenario'.

*We wanted to refer to limitation. We removed the sentences.*

P.13 29-30: "aerosol layer height is also important to determine AMF". I agree but since no analysis w.r.t this parameter are given before, it is quite hard to understand why the authors write this here...Please clarify.

*We made the sentence clearer from analysis of aerosol profiles in Fig 5.*

P. 14, 1-11: Please check what is really useful for the conclusion, and not redundant with the general part also present in the introduction. For example, it is not necessary here to repeat the nature of HCHO, why sun-synchroneous satellites are limited etc... "constellation of geostationary": first time this notion is introduced. Could you please precise it?

*We removed the first paragraph in summary of the revised manuscript. "Constellation of geostationary" was meant as GEMS, TEMPO, and Sentinel-4.*

P.14, 19: Would the ratio of hourly AMF to monthly AMF not be more useful (than the ratio of monthly to hourly) to illustrate the variability into HCHO VCDs?

*The ratio of monthly AMF to hourly AMF is more intuitive because HCHO VCDs are*

*inversely proportional to AMF. The ratio of HCHO VCDs using hourly AMF to those using monthly AMF is the same as the ratio of monthly AMF to hourly AMF.*

P.14, 32-33: "Our test with the OMI products indicated a possibility that simultaneously measured aerosol products can be used to calculate AMF considering aerosol".
This was illustrated based on the OMI AOT and SSA in the UV, but not about the aerosol layer height. Any future expectations regarding this last variable?

*Aerosol layer height can be retrieved by using O2-O2 collision (Park et al., 2016). We expect the variable can be used for geostationary satellites. We removed the lines and referred to the last in Sect. 5 as follows:*

**We only consider AOD and SSA on the AMF calculation although an aerosol layer height affects AMF calculation, which is not readily available from OMI yet. However, Park et al. (2016) recently show a possibility to retrieve aerosol height information using $O_2$-$O_2$ collision from GEMS measurements. For GEMS, we could use the retrieved aerosol information to compute scene-dependent AMFs, which will be used to improve the gas-species retrieval at each measurement time.**

P14, 8-10: The authors mentioned the importance of aerosol height in the boundary layer and to use simultaneous measurements. But no measurements about aerosols in the boundary layer are shown and used here. Where could it come from? Are such measurements available somewhere?

*You seemed to refer to P.15, 8-10. We removed the lines and discussed them in Sect. 5. Please see the paragraph above.*

P21, Figure 1: Did you compute and use the vertical averaging kernel to convert the GEOS-Chen trace gas profile into vertical column densities in order to validate your retrievals? How do you compute them and where should they be present in your OSSE diagram?

*We did not compute and use the vertical averaging kernel to convert the GEOS-Chem trace gas profile into vertical column densities because a priori profile used for AMF calculation came from GEOS-Chem and a priori profile reflects true states (GEOS-Chem simulation) in the OSSE.*

P23, Figure 3: Could you please also times that are available from geostationary observations but not from sensors like OMI (i.e. early in the morning, late in the afternoon)?

*Yes, we added 9 and 18 LST which are available time for GEMS and not for OMI in the Fig. 3-5 of the revised manuscript. Please see Fig. 3-5 above.*

P24, Figure 4: please indicate for which time(s) of the day are plotted these retrievals.

*We also added results at 9 and 18 LST. Please see the Fig. 4 above.*

P25, Figure 5: The sign of the absolute and relative differences are opposite, and thus the colours are reversed between the columns (i.e. what is red on the left, in absolute, becomes blue on the right in relative...). Please correct this.

*We intentionally plotted opposite sign. In case of relative difference between hourly and monthly AMF, the ratio of monthly to hourly AMF intuitively represents HCHO changes using hourly AMF compared to those using monthly AMF because HCHO VCDs is inversely proportional to AMF. We clarified this in the revised manuscript.*

**We also calculate percentage differences for the ratio of $AMF_m$ to $AMF_h$ at 12 LST (4[th] column in Fig. 5), which indicates changes of HCHO VCDs with $AMF_h$ relative to those with $AMF_m$ because HCHO VCDs are inversely proportional to AMF. Therefore, the percentage differences show an opposite sign from the differences between $AMF_h$ and $AMF_m$.**

P28, Figure 8: The ratio of the 2 AMFs is not strictly equal to the ratio of the 2 VDCs, since these last variables include artefacts due to the spectral fit when deriving the slant column densities. However, it represents the part of AMF computation errors included in the VDC products at the end. Please correct your second statement, in the caption, accordingly.

*Following your comments, we removed the second statement in the caption of Fig. 8.*

*We re-plotted Fig. 8 as (a) differences between AMF with and without aerosols and (b) differences between monthly average of hourly AMF and monthly AMF. We think difference is better explanation for AMF change due to aerosol effects and temporal variation effect. The ratio is clearer to explain HCHO VCD changes than difference. For reference, we updated AMF table as a function of solar zenith angles and viewing zenith angles, so values in Fig. 9 are changed. Please see our answer for AMF table in P. 12.*
*We rewrote paragraphs related to Fig. 8 and 9 in the revised manuscript as follows:*

**We calculate scene-dependent AMFs by using the OMI aerosol products together with our AMF look-up table. Figure 8(a) shows differences between monthly mean AMF with and without aerosols. AMF values with aerosols at each measurement time are calculated by using AOD and SSA from OMI. AMF values considering aerosols are higher than those without aerosols by 0.19 in absolute value, reflecting the decrement of HCHO VCDs by 11% in comparison with those without aerosols. In order to examine aerosol temporal variation effects on AMF calculation, we use the same AMF specifications discussed in Sect. 4. In the section, $AMF_h$ denotes AMF using aerosol optical properties at each measurement time, and $AMF_m$ is AMF using monthly mean AOD and SSA.**

**Figure 8(b) represents differences between monthly mean $AMF_h$ and $AMF_m$, which reflect the non-linear response of the AMF calculation due to aerosol temporal variation. Negative values are generally seen in the south of 40°N, indicating that**

**monthly mean AMF$_h$ is lower than AMF$_m$ so that HCHO column concentrations using AMF$_h$ are higher than those with AMF$_m$. The opposite sign occurs in the north of 40°N and some parts of China.**

[Figure]

**Figure 8. (a) Differences between AMFs with (AMF$_a$) and without (AMF$_{no}$) aerosols. (b) Differences of the monthly mean of AMF$_h$ versus AMF$_m$. AMF$_h$ denotes a value using AOD and SSA at each measurement time, and AMF$_m$ is a value using monthly mean AOD and SSA. Aerosol optical properties used in the calculation are from OMI observations (OMAERUV) for March 2006.**

**Finally, we examine a dust storm event on 23-29 March 2006 in order to explore an episodic case with very high aerosol concentrations. AOD and SSA (1$^{st}$ and 2$^{nd}$ rows in Fig. 9) are high and relatively low, respectively, corresponding to dust aerosols transported from the Taklamakan and Gobi deserts. As expected, the ratio of AMF without (AMF$_{no}$) to with aerosols (AMF$_a$) increases during the dust storm (3$^{rd}$ row of Fig. 9). It is a consequence of the absorbing dust aerosols transported by the dust storm. The effects are pronounced over central and northeastern China and are sometimes extended to downwind regions of Korea and the East Sea between Korea and Japan on 25 and 27 March. The ratio also increases due to biomass burning in the Indochina peninsula. The aerosol effects on AMF make HCHO VCDs increase by 32% due to absorbing aerosols and decrease by 25% due to scattering aerosols compared to those using AMF without aerosols.**

**Here we illustrate that the temporal variation effects of AOD and SSA on the AMF calculation (4$^{th}$ row in Fig. 9) can adequately be accounted for using satellite observations especially for episodic events such as dust storms and biomass burning.**

AMF$_m$ uses OMI monthly mean AOD and SSA for March 2006, and AMF$_h$ uses them at each measurement time. The ratio of AMF$_m$ to AMF$_h$ ranges from 0.68 to 1.47 reflecting HCHO changes of -32% to 47% by using AMF$_h$ compared to VCDs with AMF$_m$. That indicates that aerosol optical properties simultaneously measured for geostationary satellites can be used to calculate AMF for HCHO VCDs and to reduce the associated uncertainty with the retrieved products.

[Figure]

**Figure 9. Values of AOD, SSA, aerosol optical property effects on AMF (AMF$_{no}$/AMF$_a$), and temporal effects of aerosol optical properties on AMF (AMF$_m$/AMF$_h$) for March 23-29, 2006, when a strong dust event occurred in East Asia. AMF$_{no}$ and AMF$_a$ indicate values without and with aerosols, respectively. AMF$_m$ is a value using monthly mean AOD and SSA from OMI. AMF$_h$ is a value using AOD and SSA from OMI at each measurement time.**

**References**

Park, S. S., Kim, J., Lee, H., Torres, O., Lee, K. M., and Lee, S. D.: Utilization of O4 slant column density to derive aerosol layer height from a space-borne UV–visible hyperspectral sensor: sensitivity and case study, Atmos. Chem. Phys., 16, 1987-2006, 10.5194/acp-16-1987-2016, 2016.

---

## Referee Report (RR1)

**Review 2**

The authors have substantially modified and improved the original manuscript addressing all my questions and comments. In particular, I am satisfied now to see a better distinction between aerosol optical properties and their vertical distribution impacts on HCHO AMF variability. I thank them for these works.

I have some remaining more minor comments or questions listed below. They mostly concern some needs, on my side, to clarify news statements or analyses written in the updated manuscript.

**Main comments:**

1) P10 & 11: The analyses now clearly separate and investigate each important parameter affecting the variability of the HCHO AMF: HCHO profile, aerosol profile and aerosol optical properties. However, I feel that a few clarifications should be more emphasized or added to clarify the key messages here. These messages are somehow there but a bit complex to extract or properly summarize. Please, find my own deduced conclusions here below and see whether you (more or less) agree with them and how you can more emphasize in your respective section:

   - As you somehow mentioned on p10 l31, impacts of HCHO and aerosol profile are quite correlated. This makes sense as overall, we are looking at the resulting enhancement or shielding effect. What really matters at the end is the relative altitude between HCHO and aerosols, more than the absolute altitude of HCHO or aerosols themselves. Therefore, to properly take into account the variability of aerosols, not only their vertical distribution and optical properties have to be included but also the HCHO profile variability. This is confirmed by the numbers that somehow present same order of magnitude when looking at the differences between the AMFs. This should be properly emphasized.

   - I liked the Figure S1 that you show in your answer (AOD profiles over Eastern China). I think it is really important. Why isn't it in your manuscript? It should be there I believe. Furthermore, following my previous remark, could you add (in black for example) the vertical HCHO profile given by your model? We need to know where HCHO tropospheric bulk is located and the importance of its variability.

2) P14, l5-6: I am a bit confused here. Perhaps I am wrong but you said that the ratio of AMF without aerosol to AMF with aerosols increases. But on Fig9 3$^{rd}$ row, we clearly see that most of the area is blue (thus values below 1). Thus, this ratio looks like decreasing for me, not increasing. Am I right?
   Moreover, Figure 8 shows that the difference AMF aerosol − AMF no aerosol is higher than 0 (red colour). So that confirms that AMF with aerosol is larger. So why do you state or interpret the contrary? By the way, what is the added value here of this Fig8 compared to the Fig9? They both depict the same messages no? I think it is better to keep only the figures focusing on your selected case study (*i.e.* the days of dust storm).

P14, l10-11: I do not understand how you can see that absorbing aerosols increase VCDs while scattering aerosols decrease VCDs. Which figure does show this? Such a conclusion is not that clearly visible for me on Fig.9 Moreover, what would be the reason from physics point of view. As discussed in my former comments, and as shown in your analyses in the previous section, the key factor that determines shielding or enhancement effect is the relative altitude of HCH – aerosols. Whether particles are more scattering or more absorbing will mostly drive the magnitude of this shielding or enhancement effect.

Also, I think you should keep in mind that your computed AMF are based on your GEOS-Chem (average or hourly) aerosol profiles. If you take aerosol profiles from another model or from observations, they may differ and therefore change our AMF values, and perhaps even transform a shielding effect into an enhancement effect (or reciprocally)... Or do you mean, that, on average, scattering particles are usually elevated while absorbing aerosols are more located close to the surface (and thus below HCHO bulk)?

Please clarify here your statement, or provide elements supporting such a principle.

**Technical comments**

1) p2 l4: It should be precised that "aerosol hourly / daily variability uncertainty" cannot be neglected for Geostationary, not simply aerosol variability...

2) p4 l1-2: please check sentence, it is not clear here what you exactly mean (used words are not appropriate I believe).

p4, l4: "such as HCHO": you mean HCHO profile right? Please clarify

Moreover, you should clearly mention which input parameters you investigate in terms of temporal variations (aerosols profile, optical properties, HCHO profile). So then no ambiguities are left.

3) At several places, "achived" should be changed in "achieved". Please correct it thorough the manuscript.

4) p9, l13: "AMFmh changes hourly" => "AMFmh changes every hour"

p9, l14: "to retrieved HCHO SCDs": Please change "retrieved" into "derived" (or something similar) to avoid to duplicate the word "retrieved" already further written in the same sentence...

5) p11 l12-14: Chimot *et al.*, (2016) did not specifically investigate the impact of aerosols on HCHO but on $NO_2$. However, the findings there are, I believe, similar to any trace gas in UV-Vis. Quantitatively, numbers may vary of course. Please correct then accordingly.

6) p11, l15: "by aerosol backscatter": Please be more specific like for instance "by additional scattering effects, and thus more photons sampling the upper atmospheric layers, due to the presence of aerosols in the observed scene".

7) p13 l6: "AMF look-up table is not a function of aerosol layer heights": Perhaps to avoid any confusion for the reader, you should clearly say something like "aerosol layer heights is not an explicit input parameter of the LUT as the HCHO AMF values are based on average aerosol profile given by the GEOS-Chem simulation". On the previous page, you just mention "monthly mean data" but I believe you should explicitly refer to the aerosol profiles.

---

## Author Response (AR2)

Responses to Referee's Comments

Thanks again for raising an important issue, to which we wrote our responses in blue and the revised manuscript in red.

**Review 2**

The authors have substantially modified and improved the original manuscript addressing all my questions and comments. In particular, I am satisfied now to see a better distinction between aerosol optical properties and their vertical distribution impacts on HCHO AMF variability. I thank them for these works.

I have some remaining more minor comments or questions listed below. They mostly concern some needs, on my side, to clarify news statements or analyses written in the updated manuscript.

**Main comments:**

1) P10 & 11: The analyses now clearly separate and investigate each important parameter affecting the variability of the HCHO AMF: HCHO profile, aerosol profile and aerosol optical properties. However, I feel that a few clarifications should be more emphasized or added to clarify the key messages here. These messages are somehow there but a bit complex to extract or properly summarize. Please, find my own deduced conclusions here below and see whether you (more or less) agree with them and how you can more emphasize in your respective section:

●  As you somehow mentioned on p10 l31, impacts of HCHO and aerosol profile are quite correlated. This makes sense as overall, we are looking at the resulting enhancement or shielding effect. What really matters at the end is the relative altitude between HCHO and aerosols, more than the absolute altitude of HCHO or aerosols themselves. Therefore, to properly take into account the variability of aerosols, not only their vertical distribution and optical properties have to be included but also the HCHO profile variability. This is confirmed by the numbers that somehow present same order of magnitude when looking at the differences between the AMFs. This should be properly emphasized.

*We partly agree with you that the relative vertical distribution of HCHO and aerosol is a key factor for AMF calculations, but our analysis indicates that aerosol optical properties are also important. We emphasize our argument in the revised manuscript as follows:*

In order to examine the factors for a shielding effect ($AMF_h < AMF_m$) and an enhancement effect ($AMF_h > AMF_m$) as shown in blue and red boxes in Fig. 5(a), we plot mean profiles of aerosol and HCHO averaged over the two boxes as shown in Fig. 6. First of all, we find that aerosol profiles considerably differ between monthly and hourly values especially for its peak height, whereas relatively insignificant changes exist for HCHO profiles. The shielding effect appears to be associated with the aerosol layer higher than that of HCHO (Fig. 6(a)) and the enhancement effect is due to the opposite vertical distributions of the two (Fig. 6(b)), which is consistent with the previous studies by Leitão et al. (2010) and Chimot et al. (2016).

Our analysis further reveals the importance of aerosol optical properties especially for the shielding effect shown in the blue box of Fig. 5(a). If the relative vertical distributions of aerosol and HCHO is a single crucial factor for the shielding effect, we should expect a similar magnitude of $AMF_h$ decreases relative to $AMF_m$ for the AMF sensitivity test to aerosol vertical distributions (Fig. 5(d)). In the sensitivity test, we used the same vertical profiles of aerosol (solid) and HCHO (dotted) shown in Fig. 6(a), but the resulting changes of $AMF_h$ in Fig. 5(d) are much smaller relative to the values shown in Fig. 5(c) from the sensitivity test to aerosol optical properties. This is because the sensitivity results shown in Fig. 5(d) were obtained using the monthly mean aerosol SSA (=0.95), which is higher than hourly aerosol SSA (=0.87). In other words, the shielding effect is more pronounced with an absorbing aerosol layer rather than a scattering aerosol layer aloft, which might diminish the shielding effect by increasing a photon path length within or below the aerosol layer by the multiple light

scattering (Dickerson et al., 1997).

[Figure]

Figure 5. (a) Differences between $AMF_h$ and $AMF_m$ values and relative contributions to them by the temporal changes of (b) HCHO profiles, (c) aerosol optical properties, and (d) aerosol vertical distributions. The first to third columns are results at 9, 12, and 18 LST at Seoul on 21 June 2009. The fourth column gives percentage differences for the ratio of $AMF_m$ to $AMF_h$ indicating changes of HCHO VCDs with $AMF_h$ relative to those with $AMF_m$ at 12 LST. Blue and red boxes denote regions of shielding and enhancement effects.

[Figure]

**Figure 6. (a) Mean profiles of AOD (black) and HCHO (blue) over a region with decreased AMF$_h$ relative to AMF$_m$ (blue box in Fig. 5(a)). (b) Same as in (a) but for values over a region with increased AMF$_h$ relative to AMF$_m$ (red box in Fig. 5(a)). Solid and dotted lines denote hourly and monthly values, respectively.**

- I liked the Figure S1 that you show in your answer (AOD profiles over Eastern China). I think it is really important. Why isn't it in your manuscript? It should be there I believe. Furthermore, following my previous remark, could you add (in black for example) the vertical HCHO profile given by your model? We need to know where HCHO tropospheric bulk is located and the importance of its variability.

*Following your suggestion, we added a figure showing vertical profiles of HCHO and aerosols over regions with pronounced AMF$_h$ changes. Corresponding discussion for the figure is shown in our response above.*

2) P14, l5-6: I am a bit confused here. Perhaps I am wrong but you said that the ratio of AMF without aerosol to AMF with aerosols increases. But on Fig9 3rd row, we clearly see that most of the area is blue (thus values below 1). Thus, this ratio looks like decreasing for me, not increasing. Am I right? Moreover, Figure 8 shows that the difference AMF aerosol – AMF no aerosol is higher than 0 (red colour). So that confirms that AMF with aerosol is larger. So why do you state or interpret the contrary? By the way, what is the added value here of this Fig8 compared to the Fig9? They both depict the same messages no? I think it is better to keep only the figures focusing on your selected case study (i.e. the days of dust storm).

*Yes. You are right. The ratio of AMF without aerosol to AMF with aerosol decreases over most regions of the domain. However, the ratio increases over some regions, where AOD is high and SSA is relatively low, indicating absorbing dust aerosols.*
*We clarified this in the revised sentence as follows:*

**The ratio of $AMF_{no}$ to $AMF_a$ is less than one over most regions but higher than one over regions with dust aerosols (high AOD and relatively low SSA). The increased $AMF_a$ relative to $AMF_{no}$ is a consequence of shielding effects caused by the absorbing dust aerosols.**

*Figure 9 shows a difference between $AMF_a$ and $AMF_{no}$, and Fig. 10 shows the ratio of $AMF_{no}$ to $AMF_a$, so the results shown in the two figures are consistent. In order to clarify this, we re-plotted Figure 9(a) showing the ratio of $AMF_{no}$ to $AMF_a$.*

[Figure]

**Figure 9. (a) Ratio of AMF without aerosols (AMF$_{no}$) to AMF with aerosols (AMF$_a$). (b) Differences of the monthly mean of AMF$_h$ versus AMF$_m$. AMF$_h$ denotes a value using AOD and SSA at each measurement time, and AMF$_m$ is a value using monthly mean AOD and SSA. Aerosol optical properties used in the calculation are from OMI observations (OMAERUV) for March 2006.**

*We would like to keep Figure 9 in the revised manuscript because it shows the effects of aerosol on OMI HCHO retrieval, indicating that the present OMI HCHO column concentrations might be biased high on a monthly mean basis. We think that this bias is caused by neglecting the effect of scattering aerosols on the OMI AMF calculation in East Asia. However, in an episodic case of dust storm outbreaks shown in Fig. 10, absorbing dust aerosols have an opposite effect on AMF. Therefore, we would like to keep both figures in the manuscript to make this point.*

P14, l10-11: I do not understand how you can see that absorbing aerosols increase VCDs while scattering aerosols decrease VCDs. Which figure does show this? Such a conclusion is not that clearly visible for me on Fig.9 Moreover, what would be the reason from physics point of view. As discussed in my former comments, and as shown in your analyses in the previous section, the key factor that determines shielding or enhancement effect is the relative altitude of HCHO – aerosols. Whether particles are more scattering or more absorbing will mostly drive the magnitude of this shielding or enhancement effect.

*The ratio of AMF$_{no}$ to AMF$_a$ indicates change of HCHO VCD because HCHO VCD is inversely proportional to AMF. From the ratio in Fig. 10, we showed that absorbing*

*aerosols increase VCDs vice versa. We clarified related sentences as follows:*

**The ratio indicates the change of HCHO VCDs which are inversely proportional to AMF. Therefore, the aerosol effects on AMF make HCHO VCDs increased by 32% due to absorbing aerosols and decreased by 25% due to scattering aerosols compared to those using AMF without aerosols.**

*We discussed physical reasons related with aerosol optical properties and AMF in our responses above. Also, previous studies (Martin et al., 2003; Lee et al., 2009) referred to the effects of SSA on AMF.*

Also, I think you should keep in mind that your computed AMF are based on your GEOS-Chem (average or hourly) aerosol profiles. If you take aerosol profiles from another model or from observations, they may differ and therefore change our AMF values, and perhaps even transform a shielding effect into an enhancement effect (or reciprocally)...
Or do you mean, that, on average, scattering particles are usually elevated while absorbing aerosols are more located close to the surface (and thus below HCHO bulk)?
Please clarify here your statement, or provide elements supporting such a principle.

*We think that AMF changes do not result from only relative distribution. Aerosol optical properties can lead to shielding and enhancement effects. We discussed related physics above.*

**Technical comments**

1) p2 l4: It should be precised that "aerosol hourly / daily variability uncertainty" cannot be neglected for Geostationary, not simply aerosol variability...

*We revised the sentence as follows:*

**The impact of aerosol temporal variability cannot be neglected for future geostationary observations.**

2) p4 ll-2: please check sentence, it is not clear here what you exactly mean (used words are not appropriate I believe).

*We revised the sentence as follows:*

**Here we examine the necessity of temporal AMF for geostationary satellite observations.**

p4, l4: "such as HCHO": you mean HCHO profile right? Please clarify
Moreover, you should clearly mention which input parameters you investigate in terms of temporal variations (aerosols profile, optical properties, HCHO profile). So then no ambiguities are left.

*We revised the sentence as follows:*

**We analyze the retrieval sensitivity to AMF calculated with different temporal variations of input parameters such as aerosol optical properties and vertical distributions of HCHO and aerosol.**

3) At several places, "achived" should be changed in "achieved". Please correct it thorough the manuscript.

*We cannot find the word "achived".*

4) p9, l13: "AMFmh changes hourly" => "AMFmh changes every hour"

*We corrected it.*

p9, l14: "to retrieved HCHO SCDs": Please change "retrieved" into "derived" (or something similar) to avoid to duplicate the word "retrieved" already further written in

the same sentence...

*We corrected it.*

5) p11 l12-14: Chimot et al., (2016) did not specifically investigate the impact of aerosols on HCHO but on NO2. However, the findings there are, I believe, similar to any trace gas in UV-Vis. Quantitatively, numbers may vary of course. Please correct then accordingly.

*Thanks. We missed typo. We corrected it as follows:*

**Chimot et al. (2016) suggested the enhancement (albedo) effect associated with the relative vertical distribution between an absorbing gas and aerosol.**

6) p11, l15: "by aerosol backscatter": Please be more specific like for instance "by additional scattering effects, and thus more photons sampling the upper atmospheric layers, due to the presence of aerosols in the observed scene".

*We revised the sentence as follows:*

**HCHO absorptions increase within and above aerosol layers because of an increased photon path length caused by additional aerosol scattering effects, which is referred to as an enhancement (albedo) effect (Chimot et al., 2016).**

7) p13 l6: "AMF look-up table is not a function of aerosol layer heights": Perhaps to avoid any confusion for the reader, you should clearly say something like "aerosol layer heights is not an explicit input parameter of the LUT as the HCHO AMF values are based on average aerosol profile given by the GEOS-Chem simulation". On the previous page, you just mention "monthly mean data" but I believe you should explicitly refer to the aerosol profiles.

*Thanks for your comments. We clarified the sentence as follows:*

**However, the information is not yet available from the satellites with ultraviolet and visible channels. Thus, aerosol layer heights are not an explicit input parameter of our AMF look-up table as AMF values are based on monthly averaged aerosol profiles given by the GEOS-Chem simulation.**